# LOCAL FEATURE SWAPPING FOR GENERALIZATION IN REINFORCEMENT LEARNING

**David Bertoin**
IRT Saint-Exupéry
ISAE-SUPAERO
ANITI
Toulouse, France
david.bertoin@irt-saintexupery.com

**Emmanuel Rachelson**
ISAE-SUPAERO
Université de Toulouse
ANITI
Toulouse, France
emmanuel.rachelson@isae-supaero.fr

## ABSTRACT

Over the past few years, the acceleration of computing resources and research in deep learning has led to significant practical successes in a range of tasks, including in particular in computer vision. Building on these advances, reinforcement learning has also seen a leap forward with the emergence of agents capable of making decisions directly from visual observations. Despite these successes, the over-parametrization of neural architectures leads to memorization of the data used during training and thus to a lack of generalization. Reinforcement learning agents based on visual inputs also suffer from this phenomenon by erroneously correlating rewards with unrelated visual features such as background elements. To alleviate this problem, we introduce a new regularization technique consisting of channel-consistent local permutations (CLOP) of the feature maps. The proposed permutations induce robustness to spatial correlations and help prevent overfitting behaviors in RL. We demonstrate, on the OpenAI Procgen Benchmark, that RL agents trained with the CLOP method exhibit robustness to visual changes and better generalization properties than agents trained using other state-of-the-art regularization techniques. We also demonstrate the effectiveness of CLOP as a general regularization technique in supervised learning.

## 1 INTRODUCTION

Advances made in deep learning have opened the way to many applications in computer vision such as classification, object recognition, or image segmentation. The powerful representation capabilities of deep neural networks paved the way for many successes in deep reinforcement learning with the design of agents able to take decisions directly from pixels (Mnih et al., 2013; 2015). However, the sensitivity of neural networks to the distribution of training data strongly affects their generalization abilities. Neural networks are intrinsically designed to memorize the data they are trained upon, since their fitting implies empirical risk minimization (Vapnik, 1992) (they minimize the empirical average prediction error over a large training dataset). Therefore, they are prone to prediction errors on unseen samples. RL agents also suffer from this handicap and tend to memorize training trajectories, rather than general skills and features leading to transferable policies. This phenomenon, usually known as overfitting, takes a double sense in RL. Generalization in RL implies the ability to generalize across states (as in supervised learning), but also across environments. It is only recently that several environments with different configurations for training and testing have emerged and received a lot of attention (Nichol et al., 2018; Justesen et al., 2018; Zhang et al., 2018a; Cobbe et al., 2019; 2020), shedding light on the generalization issue which remained mostly overlooked, and confirming the poor generalization ability of current algorithms.

Strategies to achieve good generalization and avoid overfitting in deep learning fall into three categories of regularization: explicit regularization (e.g., loss penalization, weight decay), implicit regularization via the architecture and optimization (e.g., dropout, batch-normalization, batch size selection, momentum, early stopping), or implicit regularization by enhancement of the input data (data augmentation). Direct application of these strategies to deep RL agents has demonstrated some improvements in agent generalization abilities in some environments, but much progress remains to

be made to integrate RL systems in real-world applications. In this work, we address the observational overfitting issue introduced in Song et al. (2019) which considers a zero-shot generalization RL setting. An agent is trained on a specific distribution of environments (for example some levels of a platform video game) and tested on similar environments sharing the same high-level goal and dynamics but with different layouts and visual attributes (background or assets). We argue that a structured stochastic permutation of features of an RL agent during training leads to state-of-the-art generalization performance for vision-based policies. We introduce an efficient regularization technique based on Channel-consistent LOcal Permutations (CLOP) of the feature maps that mitigates overfitting. We implement it as an intermediate layer in feed-forward neural networks, and demonstrate its effectiveness on several reinforcement and supervised learning problems.

This paper is organized as follows. Section 2 presents the necessary background on generalization in supervised learning and reinforcement learning. Section 3 reviews recent work in the literature that allows for a critical look at our contribution and put it in perspective. Section 4 introduces the CLOP technique and the corresponding layer. Section 5 empirically evaluates agents using CLOP against state-of-the-art generalization methods and discusses their strengths, weaknesses, and variants. Section 6 summarizes and concludes this paper.

## 2 WHAT IS GENERALIZATION?

**Generalization in supervised learning (SL).** Let $\mathcal{X}$ be an input space of descriptors and $\mathcal{Y}$ an output space of labels. A SL problem is defined by a distribution $p(x, y)$ of elements of $\mathcal{X} \times \mathcal{Y}$, and a loss $\mathcal{L}(\hat{y}, y)$ which measures how different $\hat{y} \in \mathcal{Y}$ and $y \in \mathcal{Y}$ are. Then, for a given function $f$ intended to capture the mapping from $\mathcal{X}$ to $\mathcal{Y}$ underlying the $p$ distribution, one defines the (expected) risk as $\mathcal{R}(f) = \mathbb{E}_p[\mathcal{L}(f(x), y)]$. Since the true $p$ is generally unknown, one cannot directly minimize the risk in search for the optimal $f$. Given a training set $\mathcal{S} = \{(x_i, y_i)\}_{i=1}^{n}$ of $n$ items in $\mathcal{X} \times \mathcal{Y}$ drawn i.i.d. according to $p(x, y)$, empirical risk minimization (Vapnik, 1992) seeks to minimize $\mathcal{R}_{\mathcal{S}}(f) = 1/n \sum_{i=1}^{n} [\mathcal{L}(f(x_i), y_i)]$. The ability of $f$ to generalize to unseen samples is then defined by the generalization gap $\mathcal{R}(f) - \mathcal{R}_{\mathcal{S}}(f)$, which is often evaluated by approaching $\mathcal{R}(f)$ as the empirical risk $\mathcal{R}_{\mathcal{T}}(f)$ over a test set $\mathcal{T} = \{(x_i, y_i)\}_{i=1}^{n'}$ also drawn i.i.d. from $p(x, y)$. Closing the generalization gap can be attempted through structural risk minimization, which modifies $\mathcal{R}_{\mathcal{S}}$ so as to include a *regularization* penalty into the optimization, or more generally by the introduction of inductive biases (Mitchell, 1980).

**Reinforcement learning (RL).** RL (Sutton & Barto, 2018) considers the problem of learning a decision making policy for an agent interacting over multiple time steps with a dynamic environment. At each time step, the agent and environment are described through a state $s \in \mathcal{S}$, and an action $a \in \mathcal{A}$ is performed; then the system transitions to a new state $s'$ according to probability $T(s'|s, a)$, while receiving reward $R(s, a)$. The tuple $M = (\mathcal{S}, \mathcal{A}, T, R)$ forms a Markov Decision Process (Puterman, 2014, MDP), which is often complemented with the knowledge of an initial state distribution $p_0(s)$. A decision making policy parameterized by $\theta$ is a function $\pi_\theta(a|s)$ mapping states to distributions over actions. Training a reinforcement learning agent consists in finding the policy that maximizes the discounted expected return: $J(\pi_\theta) = \mathbb{E}[\sum_{t=0}^{\infty} \gamma^t R(s_t, a_t)]$.

**Generalization in RL.** Departing from the rather intuitive definition of generalization in SL, the idea of generalization in RL may lead to misconceptions. One could expect a policy that generalizes well to perform well *across environments*. It seems important to disambiguate this notion and stress out the difference between generalization in the SL sense (which is a single MDP problem), and domain generalization or robustness. For instance, one could expect a policy that learned to play a certain level of a platform game to be able to play on another level, provided the key game features (e.g. platforms, enemies, treasures) remain similar enough and the game dynamics are the same. This type of generalization benchmark is typically captured by procedurally generated environments, such as Procgen (Cobbe et al., 2020). Finding a policy that yields a guaranteed minimal performance among a set of MDPs (sharing common state and action spaces) is the problem of solving Robust MDPs (Iyengar, 2005). Similarly, that of policy optimization over a distribution over MDPs (sharing common states and actions too) is that of domain generalization (Tobin et al., 2017). We argue that these problems are unrelated and much harder than vanilla generalization in RL. Robust MDPs induce a $\max_\pi \min_{T,R}$ problem, domain generalization is a $\max_\pi \mathbb{E}_{T,R}$ one, while vanilla generalization as understood in SL remains a $\max_\pi$ problem that we try to solve *for*

*all states* within a single MDP (and not only the ones in the training set). Consequently, the ability to generalize in RL is a broader notion than structural risk minimization, which, unfortunately, uses the same name of generalization. In a given game, finding a policy that plays well on all levels is still solving the very same MDP. The underlying transition and reward models are the same and only the parts of the state space explored in each level are different. Note that these explored subsets of the state space may have non-empty intersections (e.g. the final part of two different levels might be common) and the optimal distribution over actions for a given state is unique. In this work we focus on generalization as the problem of preventing overfitting (and thus reducing the generalization gap) within a single MDP, which we call *observational overfitting* (Song et al., 2019).

**Observational overfitting (OO) in RL.** The inability to generalize might be caused by overfitting to the partially explored environment dynamics (Rajeswaran et al., 2017), or to some misleading signal that is correlated with progress but does not generalize to new levels (Machado et al., 2018; Song et al., 2019). Therefore, preventing OO in RL remains the ability to generalize across states, within the same MDP. Beyond the ability of RL agents to memorize good actions in explored states, it boils down to their ability to capture rules that extend to unencountered states, just as in structural risk minimization. As in SL, many policies might fit the observed data, but few might generalize to the true mechanisms of the underlying MDP. Song et al. (2019) propose to capture OO through a framework where a unique latent state space is transformed into a variety of observation spaces. Observation functions are built by combining useful features with purely decorative and unimportant ones which vary from one observation function to the next. They suppose the observation functions are drawn from a certain distribution (over procedural generation parameters for instance) and define the corresponding distribution over MDPs. In turn, they define the risk and the generalization gap with respect to this distribution over MDPs. We slightly depart from their derivation and argue this distinction is unnecessary: what is being solved is the unique MDP defined by the underlying dynamics and the projection of latent states into observations. The risk is thus defined with respect to the distribution over observations induced by the distributions over initial states and observation functions. Overall, this allows capturing the problem of generalization gap minimization in RL and underpins the developments proposed in the remainder of this paper.

## 3 RELATED WORK

In the following paragraphs, we cover essential works which aim to improve the generalization abilities of neural networks in both supervised learning and reinforcement learning.

In supervised learning, the process of modifying a learning algorithm with the objective to reduce its test error while preserving its train error is known as *regularization* (Goodfellow et al., 2016). Direct, or explicit, regularization can be achieved by adding a regularizer term into the loss function, such as an L2 penalty on networks parameters (Plaut et al., 1986; Krogh & Hertz, 1992). A second, implicit, regularization strategy consists in feature-level manipulations like dropout (Srivastava et al., 2014), drop-connect (Wan et al., 2013) or batch-normalization (Ioffe & Szegedy, 2015). Regularization can also be achieved implicitly by directly augmenting the training data with perturbations such as adding Gaussian noise (Bishop, 1995), or, in the case of visual inputs, random cropping and flipping (Krizhevsky et al., 2012; Szegedy et al., 2017) or removing structured parts of the image (DeVries & Taylor, 2017). Another efficient augmentation strategy, called label smoothing (Szegedy et al., 2016), consists in penalizing overconfident predictions of neural networks by perturbing the labels. Combining perturbation of both inputs and outputs, Zhang et al. (2018c); Yun et al. (2019), produce synthetic data and labels using two different samples and their corresponding labels.

Many studies have highlighted the limited ability of RL agents to generalize to new scenarios (Farebrother et al., 2018; Packer et al., 2018; Zhang et al., 2018b; Song et al., 2019; Cobbe et al., 2019). Using saliency maps, Song et al. (2019) exhibit that RL agents trained from pixels in environments with rich and textured observations, such as platform video games (e.g. Sonic (Nichol et al., 2018)), focus on elements of the scenery correlated with in-game progress but which lead to poor generalization in later situations. One of the identified counter-measures to overfitting in RL consists in applying standard methods in supervised learning. Cobbe et al. (2019; 2020) demonstrated the contribution of classical supervised learning regularization techniques like weight decay, dropout, or batch-normalization to generalization in procedurally generated environments. Similar to data-augmentation in supervised learning, Raileanu et al. (2020); Laskin et al. (2020); Yarats et al. (2020)

apply visual-data-augmentation on the observations provided by the environment to train robust agents. Igl et al. (2019) use a selective noise injection and information bottleneck to regularize their agent. Wang et al. (2020) propose mixreg, a direct application of mixup (Zhang et al., 2018c) in RL, combining two randomly sampled observations, and training the RL agent using their interpolated supervision signal. Lee et al. (2020) use a random convolution layer ahead of the network architecture, to modify the color and texture of the visual observations during training. Tobin et al. (2017) tackle the sim-to-real problem, using domain randomization on visual inputs to bridge the gap between simulation and reality in robotics. Raileanu & Fergus (2021) dissociate the optimization process of the policy and value function represented by separate networks and introduce an auxiliary loss that encourages the representation to be invariant to task-irrelevant properties of the environment. Another recent strategy consists in learning representations that are invariant to visual changes. Higgins et al. (2017) use a two-stage learning process: first they extract disentangled representations from random observation and then they exploit these representations to train an RL agent. Zhang et al. (2020) use bisimulation metrics to quantify behavioral similarity between states and learn robust task-relevant representations. Wang et al. (2021b) extract, without supervision, the visual foreground to provide background invariant inputs to the policy learner.

Our work focuses on the OO situations, where an agent overfits visual observations features that are irrelevant to the latent dynamics of the MDP. Our method approaches the augmentation of data by noise injection at the feature level, thus avoiding the computationally costly operations of high-dimensional image transformation in regular data augmentation. By directly modifying the encountered features' spatial localization, the CLOP layer aims to remove the correlations between spurious features and rewards.

## 4 CHANNEL-CONSISTENT LOCAL PERMUTATION LAYER

Three key intuitions underpin the proposal of the channel-consistent local permutation (CLOP) layer. 1) In many image-based decision making problems, **the important information is often spatially scarce and very localized**. The example of platform games is typical: most of the pixels are decorrelated from the MDP's objective. But this also applies to other games (e.g., driving, shooting) or environments such as image-based object manipulation. Consequently, most pixels don't convey useful information for an optimal policy and data augmentation should help disambiguate between informative and uninformative pixels. Specifically, we can expect that an image and its altered counterpart share the same reward value for the same action. Even if perturbing an image might mistakenly associate the wrong action to it, in most cases we expect noise injection in images to be beneficial for generalization since it creates synthetic images where most of the perturbed features were unimportant in the first place.
2) We refer to the process of modifying a $C \times H \times W$ image as data augmentation, whether this image is an input image or a set of feature maps produced by the $C$ filters of a convolutional layer. We argue that **data augmentation in the latent space is more efficient for generalization in decision making than in the input space**. For example, it seems more useful for generalization to learn a rule stating that an enemy sprite's position is important for the action choice, than to learn this same rule based on independent pixels, which might lead to OO. The successive convolutional layers provide greater levels of abstraction on each image: the deepest layer is the highest level of abstraction and forms a latent description space of the input images, with abstract features, like whole-object positions. Consequently, to achieve generalization in RL, data augmentation should probably be done in the latent space, on the feature maps after the deepest convolutional layer, rather than in the input space.
3) **Good actions in a state probably remain good actions in closeby states in the latent space**. For example, if one has determined that jumping is a good action in a specific image, then it is likely that if the image's objects move very locally, jumping remains a good action in the corresponding new images. This idea is related to that of Rachelson & Lagoudakis (2010) who study the locality of action domination in the state space. Consequently, we conjecture that local shifts of features in the deepest feature maps (hence representing high-level concepts and objects, rather than raw pixels) might help generate synthetic samples that help the optimization process disambiguate between useful and unimportant information. In turn, we expect this data augmentation to help prevent the bare memorization of good trajectories and instead generalize to good decision making rules.

**input:** $X$ of shape $(N, C, H, W)$
**if** training mode **then**
    $P = \text{shuffle}(\{(h, w)\}_{h \in [1, H], w \in [1, W]})$
    **for** $(h, w)$ in $P$
        $(h', w') \leftarrow$ draw a direct neighbor of $(h, w)$
        With proba $\alpha$, $\text{swap}(X[:, :, h, w], X[:, :, h', w'])$
**return** $X$

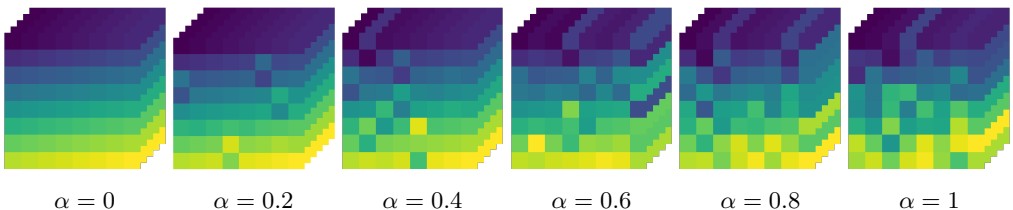

Figure 1: Channel-consistent LOcal Permutation Layer (CLOP)

$\alpha = 0$       $\alpha = 0.2$       $\alpha = 0.4$       $\alpha = 0.6$       $\alpha = 0.8$       $\alpha = 1$

Figure 2: Examples of CLOP layer outputs with different values for $\alpha$

The CLOP technique transposes these intuitions into convolutional networks. We implement it as a regularization layer that we introduce after the deepest convolutional layer's feature maps. During training, it swaps the position of pixels in these feature maps, while preserving the consistency across channels, so as to preserve the positional consistency between feature maps. This swapping is performed sequentially in a random order among pixels, locally (each pixel is swapped with one of its neighbors), and with probability $\alpha$. At evaluation time, the CLOP layer behaves like the identity function. Figure 1 illustrates this process and presents the pseudo-code. Since pixels are taken in a random order, a given pixel value might be "transported" far from its original position, with a small probability. Figure 2 illustrates the effect of varying values of $\alpha$ on the output of the layer. Since CLOP is applied after the deepest convolutional layer, each of the pixels in this image should be interpreted as a salient high-level feature in the input image, and the channels correspond to different filters. CLOP shuffles these descriptors while preserving channel consistency (no new colors are created in Figure 2) and neighborhoods (objects remain close to each other).

## 5 EXPERIMENTAL RESULTS AND DISCUSSION

This section first exhibits the CLOP method's effect on regularization in supervised learning and then demonstrates its efficiency in reinforcement learning. Hyperparameters, network architectures, and implementation choices are summarized in Appendix A and B.

### 5.1 SUPERVISED LEARNING

To assess the contribution of the CLOP layer in supervised learning, we first train a simple network (three convolutional layers followed by three linear layers) on the MNIST dataset (LeCun et al., 1998) and evaluate its generalization performance on the USPS (Hull, 1994) dataset, a slightly different digit classification dataset. Then, to confirm this performance evaluation on large-scale images, we train a VGG11 network (Simonyan & Zisserman, 2015) using the Imagenette dataset,[1] a subset of ten classes taken from Imagenet (Deng et al., 2009). In both experiments, the networks are trained using four configurations: with dropout in the dense part of the network, with batch-normalization between convolutions, with a CLOP layer after the last convolutional layer, and without any regularization. Results reported in Table 1 show that the CLOP layer allows the networks to generalize considerably better on USPS data than the unregularized network or when using dropout or batch-normalization. Note that testing on USPS implies both generalization as defined in Section 2, and *domain generalization* (Wang et al., 2021a), that is the ability to generalize to other input data distributions at test time. Likewise, applying the CLOP layer to a VGG11 improves gen-

---

[1]https://github.com/fastai/imagenette

| Method | MNIST(train) | USPS(test) | Imagenette2 (train) | Imagenette2 (test) |
|--------|--------------|------------|---------------------|--------------------|
| Plain network | $100 \pm 0.0$ | $81.6 \pm 1.2$ | $100 \pm 0.0$ | $74.7 \pm 0.7$ |
| Dropout | $99.7 \pm 0.0$ | $77.7 \pm 1.4$ | $100 \pm 0.0$ | $81.4 \pm 0.6$ |
| Batch-norm | $100 \pm 0.0$ | $65.6 \pm 7.7$ | $100 \pm 0.0$ | $82.4 \pm 0.5$ |
| CLOP (ours) | $99.9 \pm 0.0$ | $91.2 \pm 2.4$ | $100 \pm 0.0$ | $83.8 \pm 0.4$ |

Table 1: Comparison of train and test accuracies in supervised learning

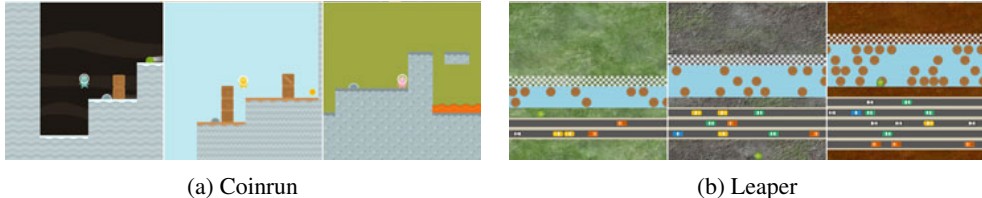

(a) Coinrun                                        (b) Leaper

Figure 3: Example of levels heterogeneity on two Procgen environments

eralization performance compared to the other methods, thus confirming the intuition that localized permutations at the feature level during training may improve generalization at testing. Appendix D features an extended discussion on the application of CLOP to other datasets (including Imagenet) and a comparison with mixup (Zhang et al., 2018c).

## 5.2 REINFORCEMENT LEARNING

We assess the regularization capability of CLOP on the Procgen benchmark, commonly used to test the generalization of RL agents. Procgen is a set of 16 visual games, each allowing procedural generation of game levels. All environments use a discrete set of 15 actions and provide $3 \times 64 \times 64$ image observations to the agent. Since all levels are procedurally generated, Procgen training and testing levels differ in many visual aspects like backgrounds, assets, or intrinsic level design (Figure 3).

**CLOP within PPO.** In the following experiments, we train all agents with PPO (Schulman et al., 2017) and the hyperparameters recommended by Cobbe et al. (2020). As in many PPO implementations, our actor and critic share a common stem of first convolutional layers, extracting shared features. Since the critic's role is only to help estimate the advantage in states that are present in the replay buffer (in other words: since the critic won't ever be used to predict values elsewhere than on the states it has been trained on), overfitting of the critic is not a critical issue in PPO. An overfitting critic will provide an oracle-like supervision signal, which is beneficial to the actor optimization process without inducing overfitting in the policy. Therefore, we choose to append a CLOP layer only to the actor part of the agent's network, located immediately after the shared feature extractor part (Appendix J discusses the application of CLOP of both actor and critic).

**Comparison with classical supervised learning regularization methods.** We first compare the performance of the data augmentation in the latent space performed by the CLOP layer with classical data augmentation methods that directly enhance inputs. Following the setup of Cobbe et al. (2020), we evaluate the zero-shot generalization of RL agents using the CLOP layer on three Procgen

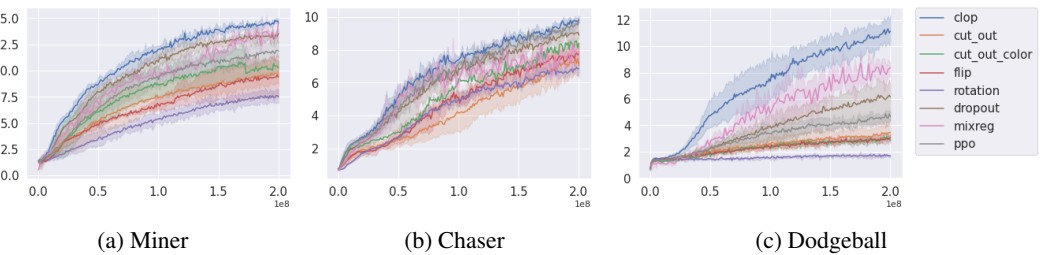

(a) Miner                    (b) Chaser                    (c) Dodgeball

Figure 4: Average sum of rewards on test environments, versus time steps.

| Game | PPO | Mixreg | Rand + FM | UCB-DraC | IBAC-SNI | RAD | IDAAC | CLOP (Ours) |
|------|-----|--------|-----------|----------|----------|-----|-------|-------------|
| Bigfish | $4.3 \pm 1.2$ | $7.1 \pm 1.6$ | $0.6 \pm 0.8$ | $9.2 \pm 2.0$ | $0.8 \pm 0.9$ | $9.9 \pm 1.7$ | $\underline{18.5 \pm 1.2}$ | $\mathbf{19.2 \pm 4.6}$ |
| BossFight | $9.1 \pm 0.1$ | $8.2 \pm 0.7$ | $1.7 \pm 0.9$ | $7.8 \pm 0.6$ | $1.0 \pm 0.7$ | $7.9 \pm 0.6$ | $\mathbf{9.8 \pm 0.6}$ | $\underline{9.7 \pm 0.1}$ |
| CaveFlyer | $5.5 \pm 0.4$ | $\underline{6.1 \pm 0.6}$ | $5.4 \pm 0.8$ | $5.0 \pm 0.8$ | $\mathbf{8.0 \pm 0.8}$ | $5.1 \pm 0.6$ | $5.0 \pm 0.6$ | $5.0 \pm 0.3$ |
| Chaser | $\underline{6.9 \pm 0.8}$ | $5.8 \pm 1.1$ | $1.4 \pm 0.7$ | $6.3 \pm 0.6$ | $1.3 \pm 0.5$ | $5.9 \pm 1.0$ | $6.8 \pm 1.0$ | $\mathbf{8.7 \pm 0.2}$ |
| Climber | $6.3 \pm 0.4$ | $6.9 \pm 0.7$ | $5.3 \pm 0.7$ | $6.3 \pm 0.6$ | $3.3 \pm 0.6$ | $6.9 \pm 0.8$ | $\mathbf{8.3 \pm 0.4}$ | $\underline{7.4 \pm 0.3}$ |
| CoinRun | $9.0 \pm 0.1$ | $8.6 \pm 0.3$ | $\underline{9.3 \pm 0.4}$ | $8.8 \pm 0.2$ | $8.7 \pm 0.6$ | $9.0 \pm 0.8$ | $\mathbf{9.4 \pm 0.1}$ | $9.1 \pm 0.1$ |
| Dodgeball | $3.3 \pm 0.4$ | $1.7 \pm 0.4$ | $0.5 \pm 0.4$ | $\underline{4.2 \pm 0.9}$ | $1.4 \pm 0.4$ | $2.8 \pm 0.7$ | $3.2 \pm 0.3$ | $\mathbf{7.2 \pm 1.2}$ |
| FruitBot | $\underline{28.5 \pm 0.2}$ | $27.3 \pm 0.8$ | $24.5 \pm 0.7$ | $27.6 \pm 0.4$ | $24.7 \pm 0.8$ | $27.3 \pm 1.8$ | $27.9 \pm 0.5$ | $\mathbf{29.8 \pm 0.3}$ |
| Heist | $2.7 \pm 0.2$ | $2.6 \pm 0.4$ | $2.4 \pm 0.6$ | $3.5 \pm 0.4$ | $\mathbf{9.8 \pm 0.6}$ | $4.1 \pm 1.0$ | $3.5 \pm 0.2$ | $\underline{4.5 \pm 0.2}$ |
| Jumper | $5.4 \pm 0.1$ | $6.0 \pm 0.3$ | $5.3 \pm 0.6$ | $6.2 \pm 0.3$ | $3.6 \pm 0.6$ | $\mathbf{6.5 \pm 0.6}$ | $\underline{6.3 \pm 0.2}$ | $5.6 \pm 0.2$ |
| Leaper | $6.5 \pm 1.1$ | $5.3 \pm 1.1$ | $6.2 \pm 0.5$ | $4.8 \pm 0.9$ | $6.8 \pm 0.6$ | $4.3 \pm 1.0$ | $\underline{7.7 \pm 1.0}$ | $\mathbf{9.2 \pm 0.2}$ |
| Maze | $5.1 \pm 0.2$ | $5.2 \pm 0.5$ | $\underline{8.0 \pm 0.7}$ | $6.3 \pm 0.1$ | $\mathbf{10.0 \pm 0.7}$ | $6.1 \pm 1.0$ | $5.6 \pm 0.3$ | $5.9 \pm 0.2$ |
| Miner | $8.4 \pm 0.4$ | $9.4 \pm 0.4$ | $7.7 \pm 0.6$ | $9.2 \pm 0.6$ | $8.0 \pm 0.6$ | $9.4 \pm 1.2$ | $\underline{9.5 \pm 0.4}$ | $\mathbf{9.8 \pm 0.3}$ |
| Ninja | $6.5 \pm 0.1$ | $6.8 \pm 0.5$ | $6.1 \pm 0.8$ | $6.6 \pm 0.4$ | $\mathbf{9.2 \pm 0.6}$ | $\underline{6.9 \pm 0.8}$ | $6.8 \pm 0.4$ | $5.8 \pm 0.4$ |
| Plunder | $6.1 \pm 0.8$ | $5.9 \pm 0.5$ | $3.0 \pm 0.6$ | $8.3 \pm 1.1$ | $2.1 \pm 0.8$ | $\underline{8.5 \pm 1.2}$ | $\mathbf{23.3 \pm 1.4}$ | $5.4 \pm 0.7$ |
| StarPilot | $36.1 \pm 1.6$ | $32.4 \pm 1.5$ | $8.8 \pm 0.7$ | $30.0 \pm 1.3$ | $4.9 \pm 0.8$ | $33.4 \pm 5.1$ | $\underline{37.0 \pm 2.3}$ | $\mathbf{40.9 \pm 1.7}$ |

Table 2: Average returns on Procgen games. Bold: best agent; underlined: second best.

environments, setting the difficulty parameter to hard. All agents were trained during 200M steps on a set of 500 training levels and evaluated on 1000 different test levels. Figure 4 shows the average sum of rewards on the test levels obtained along training for all agents: the CLOP agents outperform classical data augmentation techniques (full results and discussion in Appendix G). CLOP offers three significant advantages: asymptotic generalization performance is better, sample complexity is smaller (the agent needs fewer interactions steps to reach high performance levels) and computational complexity is negligible (see Appendix C). Interestingly, other regularization methods based on pixel-block disturbance, such as cutout or cutout-color, despite being somehow related to the CLOP layer but applied at the input image level, often even decrease the baseline performance. The CLOP layer also outperforms dropout, another feature-level manipulation method. Compared to dropout, the CLOP layer randomly shifts features from their original position instead of completely removing them. The persistence of these features in the feature map but in other positions appears to help to prevent spurious correlations with the reward signal. Overall, empirical validation confirms that the CLOP layer has a significant impact on generalization, helping prevent OO by decorrelating features that are irrelevant to the task (such as background elements) from the policy's output.

**Comparison with RL regularization methods.** We also compare the CLOP layer with current state-of-the-art regularization methods specifically designed for RL: mixreg (Wang et al., 2020), Rand+FM (Lee et al., 2020), UCB-DraC (Raileanu et al., 2020), IBAC-SNI (Igl et al., 2019), RAD (Laskin et al., 2020), and IDAAC (Raileanu & Fergus, 2021). Following the protocol proposed by Raileanu et al. (2020), we conduct a set of experiments on the easy setting of all environments available in Procgen. All agents were trained during 25M steps on 200 training levels and their performance is compared on the entire distribution of levels (Appendix G reports results on the hard setting). We report the average sum of rewards for all environments in Table 2. The CLOP layer outperforms other state-of-the-art methods on 7 out of the 16 Procgen games, sometimes by a large margin. Compared to these methods, the CLOP layer offers a direct and easy way to augment the RL agent's ability to generalize to unseen environments (Appendix E reports results on the combination with IDAAC).

**CLOP improves both performance and generalization gap.** Table 3 reports the gains obtained by the CLOP layer in terms of performance on training levels, testing levels and generalization gap, compared to PPO, after 25M steps of training on all Procgen games. Complete results are reported in Appendix F. The CLOP layer systematically improves the generalization gap (of 30% on average and up to 50% on StarPilot). It also has a notable effect on both the training and the generalization absolute performance. On all games but three (CaveFlyer, Ninja, and Plunder), the training performance is at least equivalent (10 games), and sometimes much better (BigFish, Chaser, Leaper) with the CLOP layer, both in training speed and asymptotical performance. The CLOP layer's generalization absolute performance is also better on all but the same three games,[2] with a significantly greater advantage in number of games. It reaches significantly better generalization performance on 11 games and equivalent performance on 2 games (Coinrun, Jumper).

---

[2] since the generalization gap is still reduced, we conjecture it is a matter of convergence of the training process, rather than of generalization ability

| Game | PPO train | PPO test | PPO gap | CLOP train | CLOP test | CLOP gap |
|---|---|---|---|---|---|---|
| bigfish | $18.1 \pm 4.0$ | $4.3 \pm 1.2$ | 13.8 | $+48.0\%$ | $+344.4\%$ | $-44.2\%$ |
| bossfight | $10.3 \pm 0.6$ | $9.1 \pm 0.1$ | 1.2 | $+1.9\%$ | $+6.2\%$ | $-25.0\%$ |
| caveflyer | $7.8 \pm 1.4$ | $5.5 \pm 0.4$ | 2.3 | $-15.4\%$ | $-7.9\%$ | $-30.4\%$ |
| chaser | $8.0 \pm 1.2$ | $6.9 \pm 0.8$ | 1.1 | $+18.8\%$ | $+26.7\%$ | $-18.2\%$ |
| climber | $10.2 \pm 0.4$ | $6.3 \pm 0.4$ | 3.9 | $-5.9\%$ | $+16.6\%$ | $-43.6\%$ |
| coinrun | $10.0 \pm 0.0$ | $9.0 \pm 0.1$ | 1.0 | $-0.4\%$ | $+1.5\%$ | $-20.0\%$ |
| dodgeball | $10.8 \pm 1.7$ | $3.3 \pm 0.4$ | 7.5 | $+8.2\%$ | $+117.7\%$ | $-40.0\%$ |
| fruitbot | $31.5 \pm 0.5$ | $28.5 \pm 0.2$ | 3.0 | $+0.0\%$ | $+4.3\%$ | $-40.0\%$ |
| heist | $8.8 \pm 0.3$ | $2.7 \pm 0.2$ | 6.1 | $+5.1\%$ | $+66.4\%$ | $-21.3\%$ |
| jumper | $8.9 \pm 0.4$ | $5.4 \pm 0.1$ | 3.5 | $-0.6\%$ | $+3.4\%$ | $-5.7\%$ |
| leaper | $7.1 \pm 1.6$ | $6.5 \pm 1.1$ | 0.6 | $+39.1\%$ | $+41.1\%$ | $+16.7\%$ |
| maze | $9.9 \pm 0.1$ | $5.1 \pm 0.2$ | 4.8 | $-1.5\%$ | $+14.9\%$ | $-20.8\%$ |
| miner | $12.7 \pm 0.2$ | $8.4 \pm 0.4$ | 4.3 | $+0.9\%$ | $+17.4\%$ | $-30.2\%$ |
| ninja | $9.6 \pm 0.3$ | $6.5 \pm 0.1$ | 3.1 | $-16.6\%$ | $-10.4\%$ | $-29.0\%$ |
| plunder | $8.9 \pm 1.7$ | $6.1 \pm 0.8$ | 2.8 | $-23.1\%$ | $-11.0\%$ | $-50.0\%$ |
| starpilot | $44.7 \pm 2.4$ | $36.1 \pm 1.6$ | 8.6 | $+2.1\%$ | $+13.3\%$ | $-45.3\%$ |

Table 3: Performance on training and testing levels, and generalization gap after 25M steps.

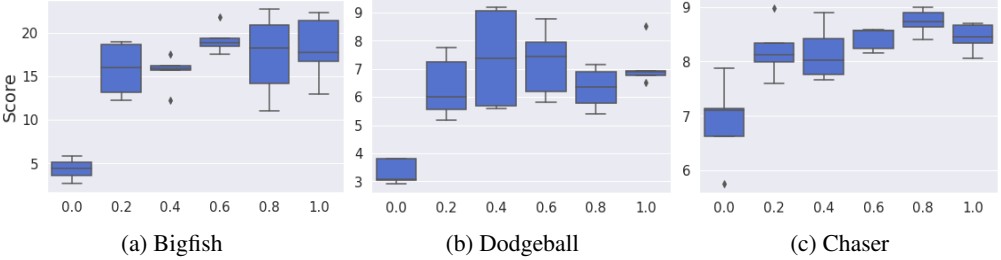

(a) Bigfish      (b) Dodgeball      (c) Chaser

Figure 5: Influence of the $\alpha$ parameter on test performance.

**Influence of the permutation rate.** The CLOP layer has a single tunable hyperparameter $\alpha$, corresponding to the probability of accepting a candidate permutation between pixels during the forward pass. Figure 5 reports the effect of varying this hyperparameter on three environments where the CLOP layer improved generalization. Interestingly, large values of $\alpha$ do not prevent the agent from learning. In some cases (Figure 5c), it might even lead to improved performance. Since permutations occur locally, a high probability of swapping does not imply the risk of generating latent states that are too far off from the original latent state (and would thus harm overall training performance). Figures 6c and 6d (see next paragraph) illustrate this risk. Another desirable property of this parameter is that performance does not seem to be too sensitive to it. Any non-zero value seems to permit generalization. Consequently, using the CLOP layer does not require fine hyperparameter tuning.

**Locality and channel-consistency are crucial features for generalization.** Figure 6 illustrates the effect of a forward pass through the CLOP layer without these components on a handcrafted, 3-channel (RGB) feature map on the Leaper game (Figure 6b). Although there is a non-zero probability that CLOP sends a pixel's content arbitrarily far from its original position, the output of the layer retains an interpretable shape from a human perspective (Figure 6c). In contrast, Figure 6d illustrates the output if the permutations are not local anymore and can send any pixel's content anywhere in the image in a single permutation. Similarly, Figure 6e shows how loosing the channel-consistency feature creates new colors in the image by recombining unrelated features together. Both these behaviors seem undesirable for proper generalization. We performed an ablation study to assess the influence of these two components. Figure 7 shows that both the locality and channel-consistency properties of CLOP are beneficial to generalization (full results in Appendix H). Interestingly, non-local permutations and channel-inconsistent permutations still provide better generalization than plain PPO. Such image modifications are agnostic to the feature maps spatial structure. But since irrelevant features (e.g., background) can be swapped without consequences, and since these features concern a majority of pixels, with high probability these modifications still generate samples that preserve the useful semantic information. However, whenever useful feature pixels are swapped without locality, this semantic information is made ambiguous between actual experience samples and augmented data, that might induce different action choices. This confirms the general idea that data augmentation helps generalization in RL, but that improved performance comes from preserv-

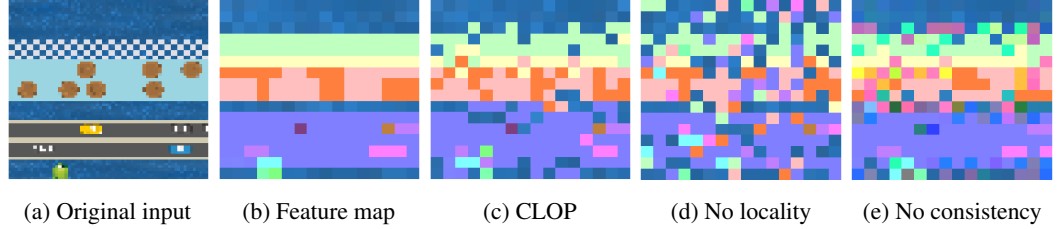

| (a) Original input | (b) Feature map | (c) CLOP | (d) No locality | (e) No consistency |

Figure 6: Applying the CLOP layer ($\alpha = 0.5$). Ablation of the locality and consistency properties.

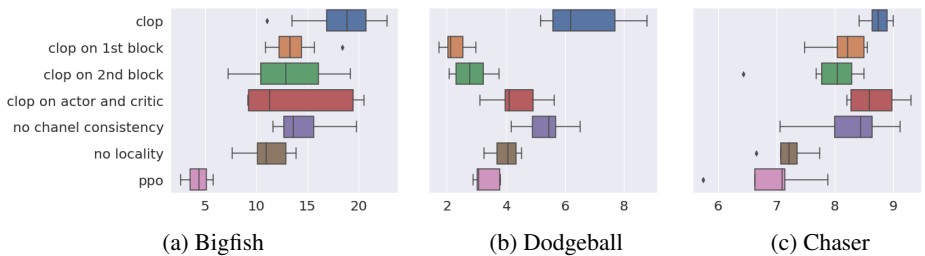

| (a) Bigfish | (b) Dodgeball | (c) Chaser |

Figure 7: Ablation study, effect on the score

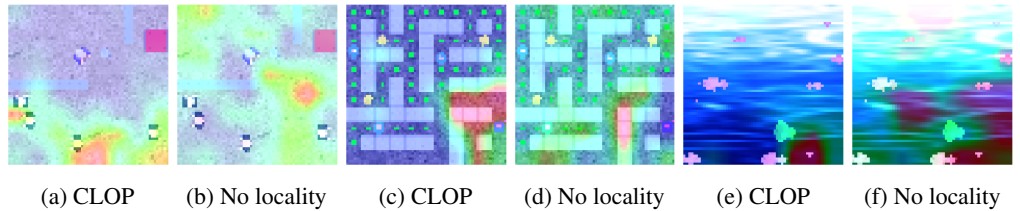

| (a) CLOP | (b) No locality | (c) CLOP | (d) No locality | (e) CLOP | (f) No locality |

Figure 8: Saliencies induced by the locality property ($\alpha = 0.5$)

ing spatial knowledge (length, width, and channel-wise) in the latent space. We also used Grad-Cam (Selvaraju et al., 2017) to produce visual explanations of the importance of these two factors (Figure 8).[3] An agent trained using a CLOP layer stripped of the local permutation feature focuses on a spread-out portion of the image, while the full CLOP agent displays very focused saliency areas. The policies of the PPO agents, or those of the CLOP agents with local but channel-inconsistent permutations, display much more focused saliency maps too, despite having lower performance than the full CLOP agents. This underlines a limit of saliency map interpretation: focussing on specific pixels is important but it is the combination of these pixel values that makes a good decision and this process is more intricate than solely concentrating on important parts of the image.

**CLOP layer's position in the network.** Figure 7 also displays the loss in generalization performance when applying CLOP to earlier layers in the IMPALA architecture used, confirming that CLOP is most beneficial on the deepest feature maps (more results in Appendix I). It also reports the effect of applying CLOP of both the actor and critic of PPO (discussion in Appendix J).

## 6 CONCLUSION

In this work, we introduce the Channel-consistent LOcal Permutations (CLOP) technique, intended to improve generalization properties of convolutional neural networks. This stems from an original perspective on observational overfitting in image-based reinforcement learning. We discuss why observational overfitting is a structural risk minimization problem that should be tackled at the level of abstract state representation features. To prevent RL agents from overfitting their decisions to in-game progress indicators and, instead, push informative, causal variables into the policy features, we

---

[3]Full videos available at `https://bit.ly/3D9fyIK`.

implement CLOP as a feed-forward layer and apply it to the deepest feature maps. This corresponds to performing data augmentation at the feature level while preserving the spatial structure of these features. The CLOP layer implies very simple operations that induce a negligible overhead cost. Agents equipped with a CLOP layer set a new state-of-the-art reference on the Procgen generalization benchmark. Although generalization in RL is a broad topic that reaches out well beyond observational overfitting, the approach taken here endeavors to shed new light and proposes a simple and low-cost original solution to this challenge, hopefully contributing a useful tool to the community.

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

# A    NETWORK ARCHITECTURE AND TRAINING HYPER-PARAMETERS

## A.1    SUPERVISED LEARNING

We used a simple network for the MNIST-USPS experiment, composed of a convolutional feature extractor:

- Conv2D(filters=32, kernel=$5 \times 5$, stride=1, padding=2, activation=ReLU)

- Max Pooling(filters=$2 \times 2$, stride=2)

- Conv2D(filters=64, kernel=$5 \times 5$, stride=1, padding=2, activation=ReLU)

- Max Pooling(filters=$2 \times 2$, stride=2)

- Conv2D(filters=128, kernel=$3 \times 3$, stride=1, padding=2, activation=ReLU)

followed by a two hidden layer MLP:

- Dense(nb_neurons=512, activation=ReLU)

- Dense(nb_neurons=100, activation=ReLU)

- Dense(nb_neurons=10, activation=Linear)

The dropout and the CLOP layers where added between the feature extractor and the classifier. For the batch-normalization configuration, we added batch-normalization layers after each convolutional layer. We trained all networks for 30 epochs with Adam, a learning rate $lr = 5 \cdot 10^{-4}$, and a cosine annealing schedule.

For the Imagenette experiment (and Imagewoof experiment of Appendix D), we used Pytorch's implementation of VGG11, replacing the output layer to match the ten classes. For the STL-10 experiment of Appendix D, since the images are already downscaled, we used Pytorch's implementation of a VGG9 network, replacing the output layer to match the ten classes. For the Imagenet experiment of Appendix D we used Pytorch's implementation of a VGG16 network, replacing the output layer to match the ten classes. When testing our method, we removed the dropout layers from the fully connected part of the network and added a CLOP layer between the last convolutional layer and the first fully connected one. As is common on these benchmarks, we applied a horizontal stochastic flipping function to preprocess the input as basic data augmentation. We trained all networks on 90 epochs with Adam, a learning rate $lr = 5 \cdot 10^{-4}$, and a cosine annealing schedule.

## A.2    REINFORCEMENT LEARNING

We based our implementation of PPO on the one proposed by Hojoon Lee available at: `https://github.com/joonleesky/train-procgen-pytorch`. We used an IMPALA (Espeholt et al., 2018) architecture on all agents, with a CLOP layer added to the actor part only for our CLOP agents (except for the ablation study reported in Figure 7 and Appendix J where it is also included in the critic). Figure 9 illustrates CLOP agent's architecture. Values of $\alpha$ used for each environment are reported in Table 4.

| Experiment | Number of epochs/steps | Time by experiment | Number of experiment repetitions |
|---|---|---|---|
| Sec. 5.1 MNIST and USPS | 30 | 2 minutes | 20 |
| Sec. 5.1 Imagenette/Imagewoof | 90 | 40 minutes | 10 |
| Sec. D STL-10 | 90 | 5 minutes | 10 |
| Sec. 5.2 Procgen Hard (for each environment) | 200M | 22 hours | 3 |
| Sec. 5.2 Procgen Easy (for each environment) | 25M | 2h40 | 5 |

Table 5: Experimental setup

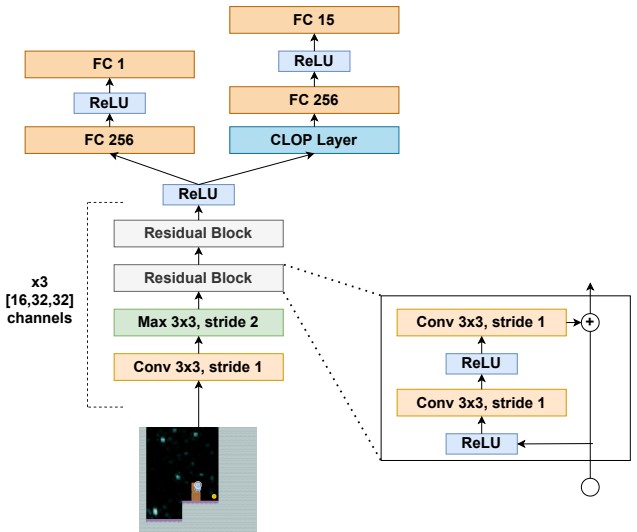

| Game | alpha |
|---|---|
| Bigfish | 0.6 |
| StarPilot | 0.3 |
| FruitBot | 0.3 |
| BossFight | 0.3 |
| Ninja | 0.3 |
| Plunder | 0.3 |
| CaveFlyer | 0.3 |
| CoinRun | 0.3 |
| Jumper | 0.3 |
| Chaser | 0.8 |
| Climber | 0.3 |
| Dodgeball | 0.6 |
| Heist | 0.3 |
| Leaper | 0.8 |
| Maze | 0.3 |
| Miner | 0.3 |

Figure 9: CLOP layer within the IMPALA architecture    Table 4: Values of $\alpha$

## B  REPRODUCIBILITY

All the experiments from Section 5 were run on a desktop machine (Intel i9, 10th generation processor, 32GB RAM) with a single NVIDIA RTX 3080 GPU. Details about each experiments are reported in Table 5.

In a general context of data efficient machine learning research, it seems relevant to underline that each subfigure in Figure 4 evaluated 7 regularization methods, over 3 repetitions. Overall, this required an order of $7 \times 3 \times 22 = 462$ computing hours (around 20 days) for a single subfigure on a high-end desktop machine. The experimental protocol used to select the three games reported in Figure 4 relies on the results obtained on the easy mode of Procgen (Figure 10 and Table 2), on a random sampling of hard mode games, and on limited training length results on hard mode games.

Besides this information, we provide the full source code of our implementation and experiments, along with the data files of experimental results we obtained.

## C  COMPUTATIONAL COST OF THE CLOP LAYER

The CLOP layer has to process each element of the intermediate feature map, but only width and height-wise. This results in a $O(H'W')$ cost, where $H'$ and $W'$ are the height and width of the feature maps, which can be much smaller than that of the input image. An important feature is the independence on the channel depth, which can grow significantly in deep layers. Comparatively, methods that perform data augmentation in the input space generally suffer from complexities at least in $O(CHW)$ (for input images of size $C \times H \times W$), and sometimes more for costly augmentation methods like rotations. Table 6 reports two examples of the computational overhead of introducing the CLOP layer in a PPO training. Table 7 performs a similar experiment comparing the running times of various standard data augmentation techniques with CLOP on MNIST and Imagenette. This confirms the negligible overhead induced by the CLOP layer.

|  | Without CLOP | With CLOP |
|---|---|---|
| Jumper easy mode | 2h 40min | 2h 41min |
| Jumper hard mode | 22h 55min | 23h 1min |
| Caveflyer easy mode | 2h 40min | 2h 41min |
| Caveflyer hard mode | 22h 25min | 22h 34min |

Table 6: Computation time with and without clop on two Progen games

|  | MNIST | Imagenette |
|---|---|---|
| No augmentation | 6.6s | 41.6s |
| Random crop | 16.8s | 41.7s |
| Rotation | 18.1s | 54.1s |
| Cutout | 6.9s | 41.7s |
| Random convolutions | N/A | 42.3s |
| CLOP | 6.6s | 41.7s |

Table 7: Computation time of one forward pass on the entire dataset

## D    ADDITIONAL EXPERIMENTS IN SUPERVISED LEARNING

Following the experiments of Section 5.1, we evaluated the benefit of the CLOP layer when training a VGG16 network on the full Imagenet database over 90 epochs, and compared with the reported improvement brought by mixup (Zhang et al., 2018c) in the same setting. Mixup improves the generalization performance by 0.2 to 0.6% across network architectures (generalization accuracies without mixup between 76 and 80% depending on the network used). Similarly, CLOP brings a negligible 0.15% improvement (71.6% top-1 generalization accuracy without CLOP, versus 71.75% with CLOP). We conjecture the sample variety and diversity in Imagenet actually cancels the need for data augmentation altogether. To confirm this conjecture and refine the understanding of when CLOP is beneficial or not, we ran the same experiment on several datasets with less variety in the images and less data. STL-10 (Coates et al., 2011) is a subset of Imagenet inpired by CIFAR-10 (Krizhevsky, 2009), where each class has even fewer labeled training examples than in CIFAR-10 (500 training images, 800 test images per class). Imagewoof is the subset of 10 breed classes from ImageNet. Training on Imagewoof or on Imagenette does not benefit from the diversity of image features present in the whole Imagenet dataset. In these cases, CLOP was the method that contributed most to close the generalization gap, as shown in Table 8. These results, along with the ones on the Imagenette dataset indicate that the CLOP layer becomes really beneficial when data is scarce and one needs to avoid overfitting.

## E    COMBINATION OF CLOP AND IDAAC

One advantage of CLOP is its simplicity and the possibility to combine it with other regularization methods. Since, at the time of writing, IDAAC is one of the most efficient algorithms on Procgen, it is tempting to evaluate its combination with CLOP. Table 9 reports on this, on all Procgen games, easy setting, after 25M training steps. CLOP was included in the IDAAC code provided in https://github.com/rraileanu/idaac.

|  | STL-10 | | Imagewoof | | Imagenette | |
|---|---|---|---|---|---|---|
|  | train | test | train | test | train | test |
| without regularization | 100 ± 0.0 | 66.8 ± 1.4 | 99.9 ± 0.1 | 51.2 ± 1.2 | 100 ± 0.0 | 74.7 ± 0.7 |
| with CLOP | 100 ± 0.0 | 72.4 ± 0.6 | 100 ± 0.0 | 71.9 ± 1.6 | 100 ± 0.0 | 83.8 ± 0.6 |
| with Mixup | 99.9 ± 0.0 | 65.1 ± 3.6 | 46.1 ± 46.0 | 25.1 ± 19.2 | 100 ± 0.0 | 76.6 ± 0.9 |
| with Dropout | 100 ± 0.0 | 70.6 ± 1.4 | 99.9 ± 0.1 | 63.2 ± 1.5 | 100 ± 0.0 | 81.4 ± 0.5 |
| with Batch norm | 100 ± 0.0 | 70.7 ± 0.6 | 99.3 ± 0.5 | 72.1 ± 1.6 | 100 ± 0.0 | 82.4 ± 0.4 |

Table 8: Training and testing accuracy (in %) on various scarce-data supervised learning tasks

| Game | IDAAC | CLOP | IDAAC + CLOP |
|---|---|---|---|
| Bigfish | $18.5 \pm 1.2$ | $19.2 \pm 4.6$ | $21.7 \pm 2.1$ |
| BossFight | $9.8 \pm 0.6$ | $9.7 \pm 0.1$ | $9.5 \pm 0.7$ |
| CaveFlyer | $5.0 \pm 0.6$ | $5.0 \pm 0.3$ | $5.0 \pm 0.4$ |
| Chaser | $6.8 \pm 1.0$ | $8.7 \pm 0.2$ | $6.0 \pm 0.5$ |
| Climber | $8.3 \pm 0.4$ | $7.4 \pm 0.3$ | $8.3 \pm 0.2$ |
| CoinRun | $9.4 \pm 0.1$ | $9.1 \pm 0.1$ | $9.2 \pm 0.5$ |
| Dodgeball | $3.2 \pm 0.3$ | $7.2 \pm 1.2$ | $3.4 \pm 0.4$ |
| FruitBot | $27.9 \pm 0.5$ | $29.8 \pm 0.3$ | $26.8 \pm 1.8$ |
| Heist | $3.5 \pm 0.2$ | $4.5 \pm 0.2$ | $3.6 \pm 0.8$ |
| Jumper | $6.3 \pm 0.2$ | $5.6 \pm 0.2$ | $5.8 \pm 0.2$ |
| Leaper | $7.7 \pm 1.0$ | $9.2 \pm 0.2$ | $6.5 \pm 2.4$ |
| Maze | $5.6 \pm 0.3$ | $5.9 \pm 0.2$ | $5.6 \pm 0.2$ |
| Miner | $9.5 \pm 0.4$ | $9.8 \pm 0.3$ | $9.9 \pm 0.1$ |
| Ninja | $6.8 \pm 0.4$ | $5.8 \pm 0.4$ | $6.4 \pm 0.5$ |
| Plunder | $23.3 \pm 1.4$ | $5.4 \pm 0.7$ | $20.3 \pm 3.2$ |
| StarPilot | $37.0 \pm 2.3$ | $40.9 \pm 1.7$ | $36.3 \pm 3.5$ |

Table 9: IDAAC+CLOP on testing levels

Overall, the combination neither seems beneficial or detrimental and the effects are rather game dependent. Being able to combine methods together has a "cocktail effect" that does not necessarily imply that the combination will be systematically more efficient than the base elements. IDAAC+CLOP substantially improves on IDAAC only in the Bigfish game. Conversely, it seems to degrade its performance on Plunder. On all other games, the improvement or degradation with respect to IDAAC seems more marginal. IDAAC's auxiliary tasks on the policy network aim at predicting the advantage and not being able to predict the ordering between frames (which is strongly connected with predicting the value). This focuses the features on essential information for the policy, thus creating an inductive bias and, maybe removing the need for data augmentation. Depending on the game, swapping pixels that contain policy-relevant information might thus have a positive or negative impact on the generalization performance. On the other hand, our implementation of CLOP is based on a standard PPO, where the policy and value networks are not separated and thus the feature maps contain both policy and value-relevant features, and thus the improvement brought by CLOP is more important than on IDAAC. This discussion also sheds light on the cases when the IDAAC features seemed insufficient to provide a good policy, i.e. when PPO+CLOP was already better than IDAAC: in these cases, adding CLOP to IDAAC cannot help recover the performance of PPO+CLOP (e.g. Chaser, Dodgeball, Fruitbot, Heist, Leaper, StarPilot).

## F    FULL RESULTS ON PROCGEN'S EASY MODE

Table 3 reported the gain in generalization gap between plain PPO and PPO+CLOP in percentages, for better readability. Specifically, it reported the gain in training accuracy, testing accuracy and generalization gap with respect those of the PPO baseline. We emphasize it is a gain in generalization gap with respect to the PPO baseline, not the generalization gap itself.

In order to assess the generalization gap itself, we calculate it as the difference between training and testing levels, as a percentage of the training performance. Table 11 reports the performance on training levels of all methods, borrowing the values mentioned in the work of Raileanu et al. (2020); Raileanu & Fergus (2021). This information is then combined with the performance on testing levels in order to assess the generalization gap of all methods, reported in Table 12. This generalization gap should be read with the absolute generalization performance in mind: a small generalization gap alone is not sufficient to characterize a method.

Figure 10 illustrates the evolution of both the performance and generalization gap along training, for PPO and PPO+CLOP.

In order to follow the protocol of (Cobbe et al., 2020) and for a comparison between CLOP and IDAAC, we also report normalized scores in Table 13, both on training and testing levels. Note that

| Game | PPO train | PPO test | PPO gap | CLOP train | CLOP test | CLOP gap |
|---|---|---|---|---|---|---|
| bigfish | $18.1 \pm 4.0$ | $4.3 \pm 1.2$ | 13.8 | $26.9 \pm 2.0$ | $19.2 \pm 1.6$ | 7.7 |
| bossfight | $10.3 \pm 0.6$ | $9.1 \pm 0.1$ | 1.2 | $10.6 \pm 0.6$ | $9.7 \pm 0.1$ | 0.9 |
| caveflyer | $7.8 \pm 1.4$ | $5.5 \pm 0.4$ | 2.3 | $6.6 \pm 0.5$ | $5.0 \pm 0.3$ | 1.6 |
| chaser | $8.0 \pm 1.2$ | $6.9 \pm 0.8$ | 1.1 | $9.6 \pm 1.0$ | $8.7 \pm 0.2$ | 0.9 |
| climber | $10.2 \pm 0.4$ | $6.3 \pm 0.4$ | 3.9 | $9.6 \pm 0.2$ | $7.4 \pm 0.3$ | 2.2 |
| coinrun | $10.0 \pm 0.0$ | $9.0 \pm 0.1$ | 1.0 | $9.9 \pm 0.1$ | $9.1 \pm 0.1$ | 0.8 |
| dodgeball | $10.8 \pm 1.7$ | $3.3 \pm 0.4$ | 7.5 | $11.7 \pm 1.5$ | $7.2 \pm 1.2$ | 4.5 |
| fruitbot | $31.5 \pm 0.5$ | $28.5 \pm 0.2$ | 3.0 | $31.6 \pm 1.0$ | $29.8 \pm 0.3$ | 1.8 |
| heist | $8.8 \pm 0.3$ | $2.7 \pm 0.2$ | 6.1 | $9.3 \pm 0.4$ | $4.5 \pm 0.2$ | 4.8 |
| jumper | $8.9 \pm 0.4$ | $5.4 \pm 0.1$ | 3.5 | $8.9 \pm 0.3$ | $5.6 \pm 0.2$ | 3.3 |
| leaper | $7.1 \pm 1.6$ | $6.5 \pm 1.1$ | 0.6 | $9.9 \pm 0.2$ | $9.2 \pm 0.2$ | 0.7 |
| maze | $9.9 \pm 0.1$ | $5.1 \pm 0.2$ | 4.8 | $9.7 \pm 0.1$ | $5.9 \pm 0.2$ | 3.8 |
| miner | $12.7 \pm 0.2$ | $8.4 \pm 0.4$ | 4.3 | $12.8 \pm 0.1$ | $9.8 \pm 0.3$ | 3.0 |
| ninja | $9.6 \pm 0.3$ | $6.5 \pm 0.1$ | 3.1 | $8.0 \pm 0.5$ | $5.8 \pm 0.4$ | 2.2 |
| plunder | $8.9 \pm 1.7$ | $6.1 \pm 0.8$ | 2.8 | $6.8 \pm 1.0$ | $5.4 \pm 0.7$ | 1.4 |
| starpilot | $44.7 \pm 2.4$ | $36.1 \pm 1.6$ | 8.6 | $45.6 \pm 2.8$ | $40.9 \pm 1.7$ | 4.7 |

Table 10: Performance on training and testing levels, and generalization gap after 25M steps

these scores should be taken with a grain of salt since the normalizing $R_{max}$ can be quite far from the actual top performance among algorithms (e.g. Starpilot, CaveFlyer, Dodgeball, BigFish). This has an effect of shrinking the apparent performance of some methods on average since it returns a (somehow artificially) low normalized score on these games. These normalized scores are multiplied by 100 for better readability. Figure 11 displays the histograms of the difference between CLOP and IDAAC on training and testing levels. The specific case of the Plunder game seems to be an outlier where IDAAC vastly outperforms CLOP (and all other methods from the literature) on testing levels (by 70.2 normalized points). For all other games, on testing levels, IDAAC dominates by 0 to 15.4 normalized points on 5 games, and CLOP dominates by 0 to 22.9 normalized points on the remaining 10 games. The average over these 15 games shows CLOP being marginally better (+3.7). Results on training levels also show an advantage for CLOP (+13.5 on average, without specific outliers). Again, we warn against over-interpretation of these figures, that depend on the normalization factors. We would also like to emphasize that these results compare the two top-performing methods (which both bring significant improvements with respect to other baselines). Given these results, it also seems relevant to note that both IDAAC and CLOP bring different insights on what is important for regularization in RL, therefore, beyond the raw comparison, these figures should mainly open new questions as to what makes a method better in each case.

Tables 14, 15 and 16 follow the outline of Table 3 and report the improvement of each method from the literature, with respect to the PPO baseline in terms of (respectively) training levels performance, testing levels performance, and generalization gap. We report them mainly for the sake of completeness.

# G   FULL RESULTS ON PROCGEN'S HARD MODE

CLOP's direct competitor, IDAAC, does not report results on Procgen's hard mode. As indicated in Appendix B, each subfigure in Figure 4 required around 20 days of computation on a high-end machine.

The experimental protocol in the Procgen paper implies testing on the full distribution of levels, including those used during training. Our choice of using only the testing levels makes the benchmark even harder since no level seen during training is among our testing distribution. We argue our protocol has two benefits. We test only the generalization property, without any influence from the performance on training levels. Also, the levels selected for testing along the training process are precisely the same across algorithms.

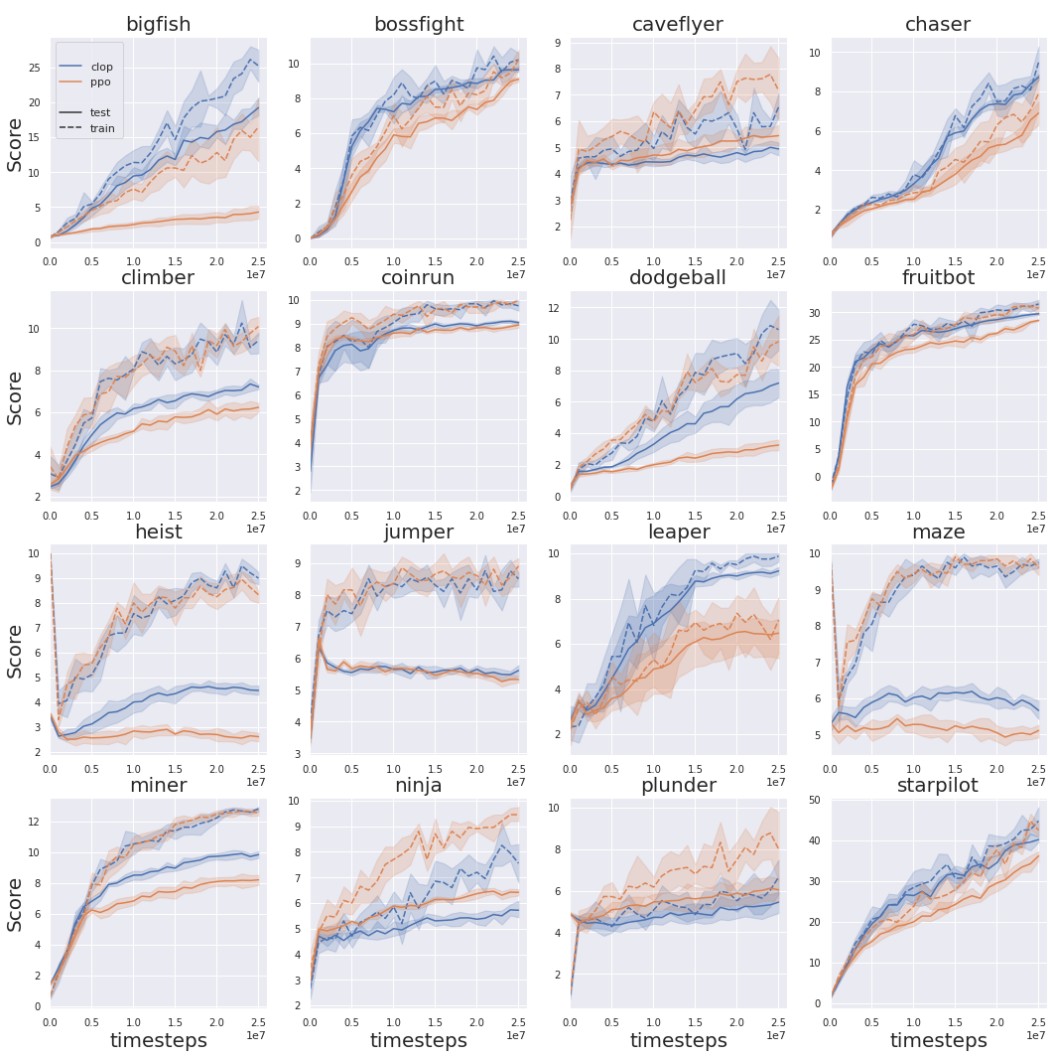

Figure 10: Train and test learning curves in easy mode

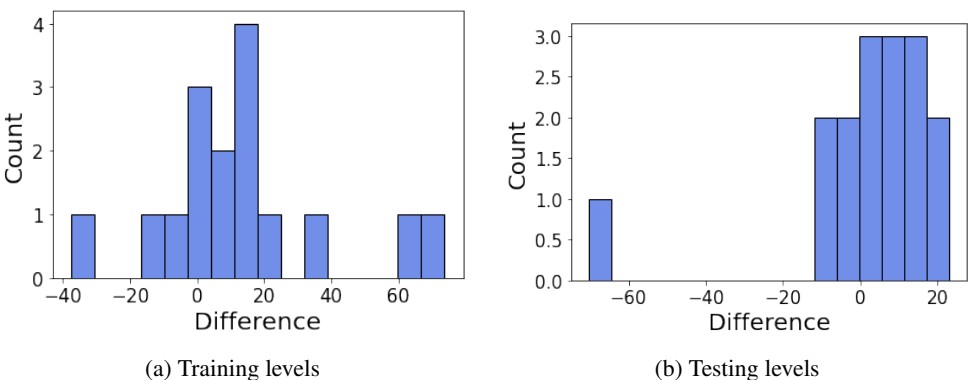

(a) Training levels

(b) Testing levels

Figure 11: Comparison of CLOP and IDAAC's performance (data from Table 13)

| Game | PPO | Mixreg | Rand + FM | UCB-DraC | IBAC-SNI | RAD | IDAAC | CLOP (Ours) |
|---|---|---|---|---|---|---|---|---|
| Bigfish | $18.1 \pm 4.0$ | $15.0 \pm 1.3$ | $6.0 \pm 0.9$ | $12.8 \pm 1.8$ | $19.1 \pm 0.8$ | $13.2 \pm 2.8$ | $\underline{21.8 \pm 1.8}$ | $\mathbf{26.9 \pm 2.0}$ |
| BossFight | $10.3 \pm 0.6$ | $7.9 \pm 0.8$ | $5.6 \pm 0.7$ | $8.1 \pm 0.4$ | $7.9 \pm 0.7$ | $8.1 \pm 1.1$ | $\underline{10.4 \pm 0.4}$ | $\mathbf{10.6 \pm 0.6}$ |
| CaveFlyer | $\mathbf{7.8 \pm 1.4}$ | $6.2 \pm 0.7$ | $6.5 \pm 0.5$ | $5.8 \pm 0.9$ | $6.2 \pm 0.5$ | $6.0 \pm 0.8$ | $6.2 \pm 0.6$ | $\underline{6.6 \pm 0.5}$ |
| Chaser | $\underline{8.0 \pm 1.2}$ | $3.4 \pm 0.9$ | $2.8 \pm 0.7$ | $7.0 \pm 0.6$ | $3.1 \pm 0.8$ | $6.4 \pm 1.0$ | $7.5 \pm 0.8$ | $\mathbf{9.6 \pm 1.0}$ |
| Climber | $\mathbf{10.2 \pm 0.4}$ | $7.5 \pm 0.8$ | $7.5 \pm 0.8$ | $8.6 \pm 0.6$ | $7.1 \pm 0.7$ | $9.3 \pm 1.1$ | $\underline{10.2 \pm 0.7}$ | $9.6 \pm 0.2$ |
| CoinRun | $\mathbf{10.0 \pm 0.0}$ | $9.5 \pm 0.2$ | $9.6 \pm 0.6$ | $9.4 \pm 0.2$ | $9.6 \pm 0.4$ | $9.6 \pm 0.4$ | $9.8 \pm 0.1$ | $\underline{9.9 \pm 0.1}$ |
| Dodgeball | $\underline{10.8 \pm 1.7}$ | $9.1 \pm 0.5$ | $4.3 \pm 0.3$ | $7.3 \pm 0.8$ | $9.4 \pm 0.6$ | $5.0 \pm 0.7$ | $4.9 \pm 0.3$ | $\mathbf{11.7 \pm 1.5}$ |
| FruitBot | $\underline{31.5 \pm 0.5}$ | $29.9 \pm 0.5$ | $29.2 \pm 0.7$ | $29.3 \pm 0.5$ | $29.4 \pm 0.8$ | $26.1 \pm 3.0$ | $29.1 \pm 0.7$ | $\mathbf{31.6 \pm 1.0}$ |
| Heist | $\underline{8.8 \pm 0.3}$ | $4.4 \pm 0.3$ | $6.0 \pm 0.5$ | $6.2 \pm 0.6$ | $4.8 \pm 0.7$ | $6.2 \pm 0.9$ | $4.5 \pm 0.3$ | $\mathbf{9.3 \pm 0.4}$ |
| Jumper | $\underline{8.9 \pm 0.4}$ | $8.5 \pm 0.4$ | $8.9 \pm 0.4$ | $8.2 \pm 0.1$ | $8.5 \pm 0.6$ | $8.6 \pm 0.4$ | $8.7 \pm 0.2$ | $\mathbf{8.9 \pm 0.3}$ |
| Leaper | $7.1 \pm 1.6$ | $3.2 \pm 1.2$ | $3.2 \pm 0.7$ | $5.0 \pm 0.9$ | $2.7 \pm 0.4$ | $4.9 \pm 0.9$ | $\underline{8.3 \pm 0.7}$ | $\mathbf{9.9 \pm 0.2}$ |
| Maze | $\mathbf{9.9 \pm 0.1}$ | $8.7 \pm 0.7$ | $8.9 \pm 0.6$ | $8.5 \pm 0.3$ | $8.2 \pm 0.8$ | $8.4 \pm 0.7$ | $6.4 \pm 0.5$ | $\underline{9.7 \pm 0.1}$ |
| Miner | $12.7 \pm 0.2$ | $8.9 \pm 0.9$ | $\underline{11.7 \pm 0.8}$ | $12.0 \pm 0.3$ | $8.5 \pm 0.7$ | $12.6 \pm 1.0$ | $11.5 \pm 0.5$ | $\mathbf{12.8 \pm 0.1}$ |
| Ninja | $\mathbf{9.6 \pm 0.3}$ | $8.2 \pm 0.4$ | $7.2 \pm 0.6$ | $8.0 \pm 0.4$ | $8.3 \pm 0.8$ | $8.9 \pm 0.9$ | $\underline{8.9 \pm 0.3}$ | $8.0 \pm 0.5$ |
| Plunder | $8.9 \pm 1.7$ | $6.2 \pm 0.3$ | $5.5 \pm 0.7$ | $\underline{10.2 \pm 1.8}$ | $6.0 \pm 0.6$ | $8.4 \pm 1.5$ | $\mathbf{24.6 \pm 1.6}$ | $6.8 \pm 1.0$ |
| StarPilot | $\underline{44.7 \pm 2.4}$ | $28.7 \pm 1.1$ | $26.3 \pm 0.8$ | $33.1 \pm 1.3$ | $26.7 \pm 0.7$ | $36.5 \pm 3.9$ | $38.6 \pm 2.2$ | $\mathbf{45.6 \pm 2.8}$ |

Table 11: Average returns on Procgen training levels. Bold: best agent; underlined: second best.

| Game | PPO | Mixreg | Rand + FM | UCB-DraC | IBAC-SNI | RAD | IDAAC | CLOP (Ours) |
|---|---|---|---|---|---|---|---|---|
| Bigfish | $-76,24\%$ | $-52,67\%$ | $-90,00\%$ | $-28,13\%$ | $-95,81\%$ | $-25,00\%$ | $-15,14\%$ | $-28,62\%$ |
| BossFight | $-11,65\%$ | $+3,80\%$ | $-69,64\%$ | $-3,70\%$ | $-87,34\%$ | $-2,47\%$ | $-5,77\%$ | $-8,49\%$ |
| CaveFlyer | $-29,49\%$ | $-1,61\%$ | $-16,92\%$ | $-13,79\%$ | $+29,03\%$ | $-15,00\%$ | $-19,35\%$ | $-24,24\%$ |
| Chaser | $-13,75\%$ | $+70,59\%$ | $-50,00\%$ | $-10,00\%$ | $-58,06\%$ | $-7,81\%$ | $-9,33\%$ | $-9,38\%$ |
| Climber | $-38,24\%$ | $-8,00\%$ | $-29,33\%$ | $-26,74\%$ | $-53,52\%$ | $-25,81\%$ | $-18,63\%$ | $-22,92\%$ |
| CoinRun | $-10,00\%$ | $-9,47\%$ | $-3,12\%$ | $-6,38\%$ | $-9,38\%$ | $-6,25\%$ | $-4,08\%$ | $-8,08\%$ |
| Dodgeball | $-69,44\%$ | $-81,32\%$ | $-88,37\%$ | $-42,47\%$ | $-85,11\%$ | $-44,00\%$ | $-34,69\%$ | $-38,46\%$ |
| FruitBot | $-9,52\%$ | $-8,70\%$ | $-16,10\%$ | $-5,80\%$ | $-15,99\%$ | $+4,60\%$ | $-4,12\%$ | $-5,70\%$ |
| Heist | $-69,32\%$ | $-40,91\%$ | $-60,00\%$ | $-43,55\%$ | $+104,17\%$ | $-33,87\%$ | $-22,22\%$ | $-51,61\%$ |
| Jumper | $-39,33\%$ | $-29,41\%$ | $-40,45\%$ | $-24,39\%$ | $-57,65\%$ | $-24,42\%$ | $-27,59\%$ | $-37,08\%$ |
| Leaper | $-8,45\%$ | $+65,63\%$ | $+93,75\%$ | $-4,00\%$ | $+151,85\%$ | $-12,24\%$ | $-7,23\%$ | $-7,07\%$ |
| Maze | $-48,48\%$ | $-40,23\%$ | $-10,11\%$ | $-25,88\%$ | $21,95\%$ | $-27,38\%$ | $-12,50\%$ | $-39,18\%$ |
| Miner | $-33,86\%$ | $+5,62\%$ | $-34,19\%$ | $-23,33\%$ | $-5,88\%$ | $-25,40\%$ | $-17,39\%$ | $-23,44\%$ |
| Ninja | $-32,29\%$ | $-17,07\%$ | $-15,28\%$ | $-17,50\%$ | $+10,84\%$ | $-22,47\%$ | $-23,60\%$ | $-27,50\%$ |
| Plunder | $-31,46\%$ | $-4,84\%$ | $-45,45\%$ | $-18,63\%$ | $-65,00\%$ | $+1,19\%$ | $-5,28\%$ | $-20,59\%$ |
| StarPilot | $-19,24\%$ | $+12,89\%$ | $-66,54\%$ | $-9,37\%$ | $-81,65\%$ | $-8,49\%$ | $-4,15\%$ | $-10,31\%$ |

Table 12: Generalization gap (difference between test and train performance)

Figure 12 shows the training curves along with the testing performance for CLOP and PPO. Overall, the main conclusions remain: CLOP dominates in a majority of games and often improves the convergence speed, both in terms of training and testing performance.

Although it was not possible to repeat the comparison of the 7 regularization methods of Figure 4 (or of other methods) on all Procgen games, mixreg (Wang et al., 2020) does report results on the hard setting. Wang et al. (2020) follow the protocol of Cobbe et al. (2020) that evaluate the agents on the full distribution of levels. Table 17 reports the compared performance on the full distribution of testing levels of PPO, mixreg, and CLOP in the hard setting, after 200M training steps. Here again, CLOP dominates (by a variable margin) mixreg in 10 out of the 16 Procgen games.

## H FULL RESULTS ON THE ABLATION STUDY

Figure 13 reports the effect of turning off the channel consistency constraint, or the locality constraint, on all Procgen games. On a majority of games (10 out of 16: Bigfish, Chaser, Climber, Dodgeball, Fruitbot, Jumper, Leaper, Maze, Miner, Starpilot), the trend shown in Section 5.2 is confirmed: both locality and channel-consistency are important for generalization performance. Three games stand-out as outliers (Bossfight, Coinrun, Heist), where turning off one of these constraints seemed to sometimes improve generalization performance. First of all, we note this improvement is small compared to the improvement of CLOP upon PPO. So this does not question the interest of using CLOP overall. Secondly, we conjecture these improvements are rather game or training-dependent. Finally, it seems illegitimate to draw conclusions on the three last games (CaveFlyer, Ninja, Plunder) since PPO outperforms CLOP on them, with or without these constraints. Interestingly, IDAAC also performs rather poorly on two games among these three (CaveFlyer and Ninja, where IBAC-SNI is the leading algorithm), while it dominates on Plunder (Table 2).

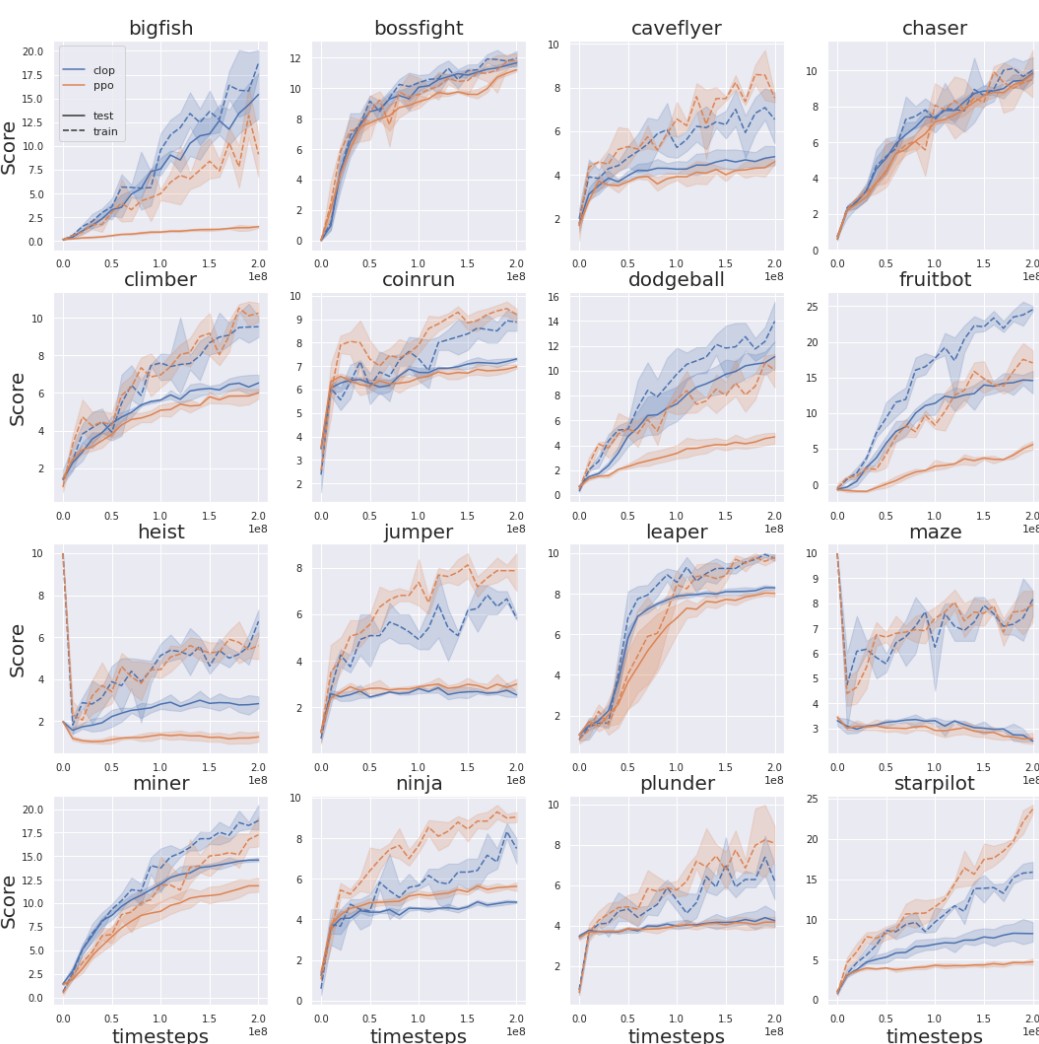

Figure 12: CLOP and PPO performance in hard mode.

| | Training levels | | | Testing levels | | |
|---|---|---|---|---|---|---|
| Game | IDAAC | CLOP | Difference | IDAAC | CLOP | Difference |
| bigfish | 53.3 | 66.4 | +13.1 | 44.9 | 46.7 | +1.8 |
| bossfight | 79.2 | 80.8 | +1.6 | 74.4 | 73.6 | −0.8 |
| caveflyer | 31.8 | 36.5 | +4.7 | 17.6 | 17.6 | +0.0 |
| chaser | 56.0 | 72.8 | +16.8 | 50.4 | 65.6 | +15.2 |
| climber | 77.4 | 71.7 | −5.7 | 59.4 | 50.9 | −8.5 |
| coinrun | 96.0 | 98.0 | +2.0 | 88.0 | 82.0 | −6.0 |
| dodgeball | 19.4 | 58.3 | +38.9 | 9.7 | 32.6 | +22.9 |
| fruitbot | 90.3 | 97.6 | +7.4 | 86.7 | 92.3 | +5.6 |
| heist | 15.4 | 89.2 | +73.8 | 0.0 | 15.4 | +15.4 |
| jumper | 81.4 | 84.3 | +2.9 | 47.1 | 37.1 | −10.0 |
| leaper | 75.7 | 98.6 | +22.9 | 67.1 | 88.6 | +21.4 |
| maze | 28.0 | 94.0 | +66.0 | 12.0 | 18.0 | +6.0 |
| miner | 87.0 | 98.3 | +11.3 | 69.6 | 72.2 | +2.6 |
| ninja | 83.1 | 69.2 | −13.8 | 50.8 | 35.4 | −15.4 |
| plunder | 46.7 | 9.0 | −37.6 | 73.7 | 3.5 | −70.2 |
| starpilot | 58.7 | 70.1 | +11.4 | 56.1 | 62.4 | +6.3 |

Table 13: Normalized score ($100\times$) on Procgen Easy mode (graphical representation of the same data on Figure 11)

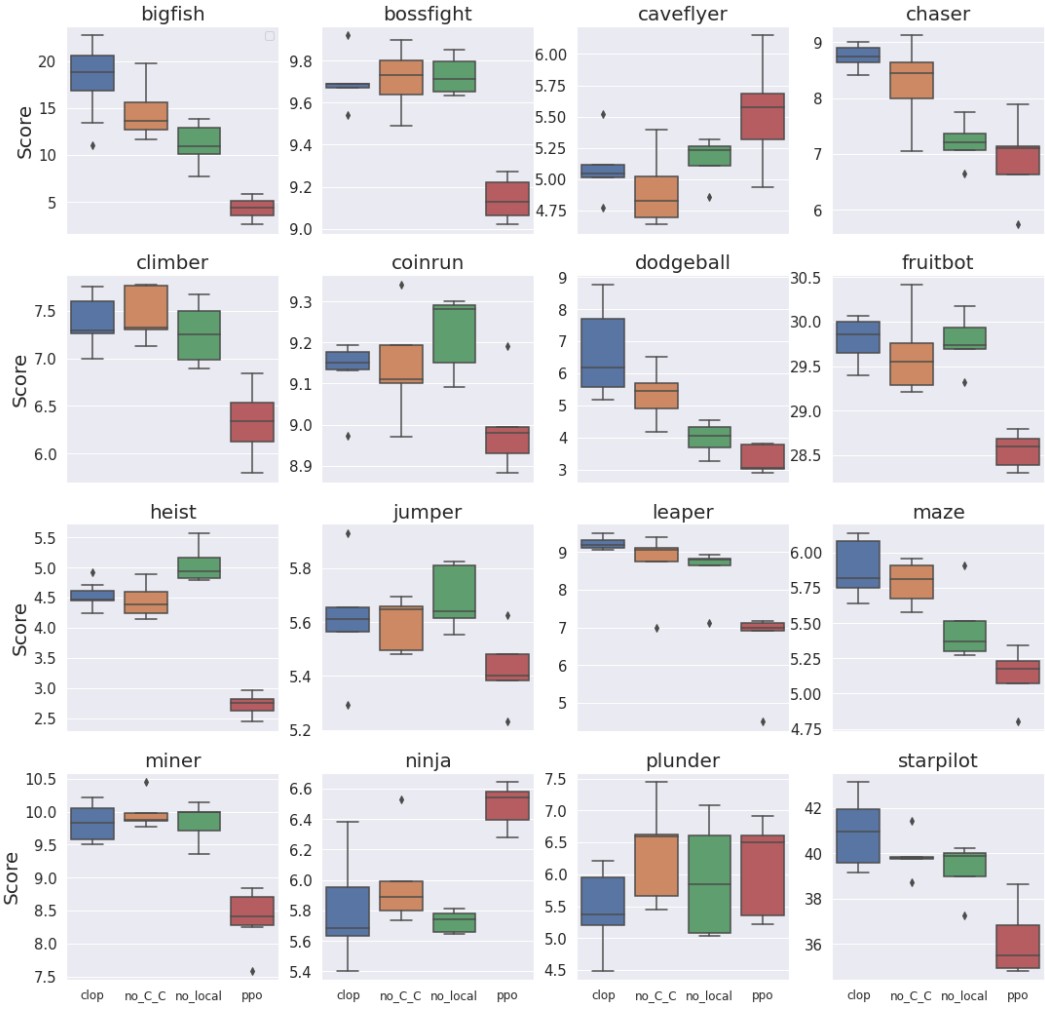

Figure 13: Ablation study on all games.

| Game | PPO | Mixreg | Rand + FM | UCB-DraC | IBAC-SNI | RAD | IDAAC | CLOP (Ours) |
|------|-----|--------|-----------|----------|----------|-----|-------|-------------|
| Bigfish | $18.1 \pm 4.0$ | $-17.1\%$ | $-66.8\%$ | $-29.3\%$ | $+5.5\%$ | $-27.1\%$ | $+20.4\%$ | $\mathbf{+48.6}\%$ |
| BossFight | $10.3 \pm 0.6$ | $-23.3\%$ | $-45.6\%$ | $-21.4\%$ | $-23.3\%$ | $-21.4\%$ | $+1.0\%$ | $\mathbf{+2.9}\%$ |
| CaveFlyer | $7.8 \pm 1.4$ | $-20.5\%$ | $-16.7\%$ | $-25.6\%$ | $-20.5\%$ | $-23.1\%$ | $-20.5\%$ | $\mathbf{-15.4}\%$ |
| Chaser | $8.0 \pm 1.2$ | $-57.5\%$ | $-65.0\%$ | $-12.5\%$ | $-61.2\%$ | $-20.0\%$ | $-6.2\%$ | $\mathbf{+20.0}\%$ |
| Climber | $10.2 \pm 0.4$ | $-26.5\%$ | $-26.5\%$ | $-15.7\%$ | $-30.4\%$ | $-8.8\%$ | $\mathbf{+0.0}\%$ | $-5.9\%$ |
| CoinRun | $10.0 \pm 0.0$ | $-5.0\%$ | $-4.0\%$ | $-6.0\%$ | $-4.0\%$ | $-4.0\%$ | $-2.0\%$ | $\mathbf{-1.0}\%$ |
| Dodgeball | $10.8 \pm 1.7$ | $-15.7\%$ | $-60.2\%$ | $-32.4\%$ | $-13.0\%$ | $-53.7\%$ | $-54.6\%$ | $\mathbf{+8.3}\%$ |
| FruitBot | $31.5 \pm 0.5$ | $-5.1\%$ | $-7.3\%$ | $-7.0\%$ | $-6.7\%$ | $-17.1\%$ | $-7.6\%$ | $\mathbf{+0.3}\%$ |
| Heist | $8.8 \pm 0.3$ | $-50.0\%$ | $-31.8\%$ | $-29.5\%$ | $-45.4\%$ | $-29.5\%$ | $-48.8\%$ | $\mathbf{+5.7}\%$ |
| Jumper | $8.9 \pm 0.4$ | $-4.5\%$ | $\mathbf{+0.0}\%$ | $-7.9\%$ | $-4.5\%$ | $-3.4\%$ | $-2.2\%$ | $\mathbf{+0.0}\%$ |
| Leaper | $7.1 \pm 1.6$ | $-54.9\%$ | $-54.9\%$ | $-29.6\%$ | $-62.0\%$ | $-31.0\%$ | $+16.9\%$ | $\mathbf{+39.4}\%$ |
| Maze | $9.9 \pm 0.1$ | $-12.1\%$ | $-10.1\%$ | $-14.1\%$ | $-17.1\%$ | $-15.1\%$ | $-35.3\%$ | $\mathbf{-2.0}\%$ |
| Miner | $12.7 \pm 0.2$ | $-29.9\%$ | $-7.9\%$ | $-5.5\%$ | $-33.1\%$ | $-0.8\%$ | $-9.4\%$ | $\mathbf{+0.8}\%$ |
| Ninja | $9.6 \pm 0.3$ | $-14.6\%$ | $-25.0\%$ | $-16.7\%$ | $-13.5\%$ | $\mathbf{-7.3}\%$ | $\mathbf{-7.3}\%$ | $-16.7\%$ |
| Plunder | $8.9 \pm 1.7$ | $-30.3\%$ | $-38.2\%$ | $+14.6\%$ | $-32.6\%$ | $-5.6\%$ | $\mathbf{+176.4}\%$ | $-23.6\%$ |
| StarPilot | $44.7 \pm 2.4$ | $-35.8\%$ | $-41.2\%$ | $-25.9\%$ | $-40.3\%$ | $-18.3\%$ | $-13.6\%$ | $\mathbf{+2.0}\%$ |

Table 14: Training levels performance versus PPO

| Game | PPO | Mixreg | Rand + FM | UCB-DraC | IBAC-SNI | RAD | IDAAC | CLOP (Ours) |
|------|-----|--------|-----------|----------|----------|-----|-------|-------------|
| Bigfish | $4.3 \pm 1.2$ | $+65.12\%$ | $-86.05\%$ | $+113.95\%$ | $-81.40\%$ | $+130.23\%$ | $+330.23\%$ | $\mathbf{+346.51}\%$ |
| BossFight | $9.1 \pm 0.1$ | $-9.89\%$ | $-81.32\%$ | $-14.29\%$ | $-89.01\%$ | $-13.19\%$ | $\mathbf{+7.69}\%$ | $+6.59\%$ |
| CaveFlyer | $5.5 \pm 0.4$ | $+10.91\%$ | $-1.82\%$ | $-9.09\%$ | $\mathbf{+45.45}\%$ | $-7.27\%$ | $-9.09\%$ | $-9.09\%$ |
| Chaser | $6.9 \pm 0.8$ | $-15.94\%$ | $-79.71\%$ | $-8.70\%$ | $-81.16\%$ | $-14.49\%$ | $-1.45\%$ | $\mathbf{+26.09}\%$ |
| Climber | $6.3 \pm 0.4$ | $+9.52\%$ | $-15.87\%$ | $+0.00\%$ | $-47.62\%$ | $+9.52\%$ | $\mathbf{+31.75}\%$ | $+17.46\%$ |
| CoinRun | $9.0 \pm 0.1$ | $-4.44\%$ | $+3.33\%$ | $-2.22\%$ | $-3.33\%$ | $+0.00\%$ | $\mathbf{+4.44}\%$ | $+1.11\%$ |
| Dodgeball | $3.3 \pm 0.4$ | $-48.48\%$ | $-84.85\%$ | $+27.27\%$ | $-57.58\%$ | $-15.15\%$ | $-3.03\%$ | $\mathbf{+118.18}\%$ |
| FruitBot | $28.5 \pm 0.2$ | $-4.21\%$ | $-14.04\%$ | $-3.16\%$ | $-13.33\%$ | $-4.21\%$ | $-2.11\%$ | $\mathbf{+4.56}\%$ |
| Heist | $2.7 \pm 0.2$ | $-3.70\%$ | $-11.11\%$ | $+29.63\%$ | $\mathbf{+262.96}\%$ | $+51.85\%$ | $+29.63\%$ | $+66.67\%$ |
| Jumper | $5.4 \pm 0.1$ | $+11.11\%$ | $-1.85\%$ | $+14.81\%$ | $-33.33\%$ | $\mathbf{+20.37}\%$ | $+16.67\%$ | $+3.70\%$ |
| Leaper | $6.5 \pm 1.1$ | $-18.46\%$ | $-4.62\%$ | $-26.15\%$ | $+4.62\%$ | $-33.85\%$ | $+18.46\%$ | $\mathbf{+41.54}\%$ |
| Maze | $5.1 \pm 0.2$ | $+1.96\%$ | $+56.86\%$ | $+23.53\%$ | $\mathbf{+96.08}\%$ | $+19.61\%$ | $+9.80\%$ | $+15.69\%$ |
| Miner | $8.4 \pm 0.4$ | $+11.90\%$ | $-8.33\%$ | $+9.52\%$ | $-4.76\%$ | $+11.90\%$ | $+13.10\%$ | $\mathbf{+16.67}\%$ |
| Ninja | $6.5 \pm 0.1$ | $+4.62\%$ | $-6.15\%$ | $+1.54\%$ | $\mathbf{+41.54}\%$ | $+6.15\%$ | $+4.62\%$ | $-10.77\%$ |
| Plunder | $6.1 \pm 0.8$ | $-3.28\%$ | $-50.82\%$ | $\mathbf{+36.07}\%$ | $-65.57\%$ | $+39.34\%$ | $+281.97\%$ | $-11.48\%$ |
| StarPilot | $36.1 \pm 1.6$ | $-10.25\%$ | $-75.62\%$ | $-16.90\%$ | $-86.43\%$ | $-7.48\%$ | $+2.49\%$ | $\mathbf{+13.30}\%$ |

Table 15: Testing levels performance versus PPO

## I  APPLYING CLOP AT VARIOUS DEPTHS WITHIN THE NETWORK

The CLOP layer can be applied after any convolutional or pooling layer of a neural network. Figure 7 shows the impact of the CLOP layer's position, after different Resnet blocks, in the IMPALA architecture, evaluated on three Procgen games. We can observe that the CLOP layer has a more substantial effect when applied on the most deepest feature map. To verify whether this is also confirmed in supervised learning, we trained two networks (architectures in Appendix A) with a CLOP layer at various depths, on MNIST and STL-10. The results, reported in Figure 14 show that the position of the CLOP layer also has an impact on supervised learning. Similarly to the phenomenon observed in RL, in supervised learning, applying the CLOP layer on the deepest features maps increases the generalization capabilities of the networks in both experiments. This goes to confirm that data augmentation over abstract features (deep convolutional feature maps) is a key to efficient generalization.

## J  APPLYING CLOP ON BOTH THE ACTOR AND THE CRITIC OF PPO

The results reported in Section 5.2 made use of CLOP only within the actor of PPO. It is legitimate to wonder whether there is any interest in applying the CLOP layer also to the critic.

It is useful to note that PPO is a "true" policy gradient method, contrarily to SAC or TD3 that are approximate value iteration methods which include gradient ascent on the policy to solve the greediness search (the $\arg\max$ operator in the Bellman equation). Because of this, it collects samples at every round to estimate the critic and cannot rely on a persistent replay buffer. Intrinsically, the critic is only used to compute advantage estimates in states that have been encountered during recent exploration. Consequently, there is no need for the critic to actually generalize to unseen states

| Game | PPO | Mixreg | Rand + FM | UCB-DraC | IBAC-SNI | RAD | IDAAC | CLOP (Ours) |
|---|---|---|---|---|---|---|---|---|
| Bigfish | 13.8 | $-42.8\%$ | $-60.9\%$ | $-73.9\%$ | $+32,6\%$ | $\mathbf{-76.1}\%$ | $\mathbf{-76.1}\%$ | $-44.2\%$ |
| BossFight | 1.2 | $-125.0\%$ | $+225.0\%$ | $-75.0\%$ | $+475,0\%$ | $\mathbf{-83.3}\%$ | $-50.0\%$ | $-25.0\%$ |
| CaveFlyer | 2.3 | $-95.7\%$ | $-52.2\%$ | $-65.2\%$ | $\mathbf{-178,3}\%$ | $-60.9\%$ | $-47.8\%$ | $-30.4\%$ |
| Chaser | 1.1 | $\mathbf{-318.2}\%$ | $+27.3\%$ | $-36.4\%$ | $+63,6\%$ | $-54.5\%$ | $-36.4\%$ | $-18.2\%$ |
| Climber | 3.9 | $\mathbf{-84.6}\%$ | $-43.6\%$ | $-41.0\%$ | $-2,6\%$ | $-38.5\%$ | $-51.3\%$ | $-43.6\%$ |
| CoinRun | 1.0 | $-10.0\%$ | $\mathbf{-70.0}\%$ | $-40.0\%$ | $-10,0\%$ | $-40.0\%$ | $-60.0\%$ | $-20.0\%$ |
| Dodgeball | 7.5 | $-1.3\%$ | $-49.3\%$ | $-58.7\%$ | $+6,7\%$ | $-70.7\%$ | $\mathbf{-77.3}\%$ | $-40.0\%$ |
| FruitBot | 3.0 | $-13.3\%$ | $+56.7\%$ | $-43.3\%$ | $+56,7\%$ | $\mathbf{-140.0}\%$ | $-60.0\%$ | $-40.0\%$ |
| Heist | 6.1 | $-70.5\%$ | $-41.0\%$ | $-55.7\%$ | $\mathbf{-182,0}\%$ | $-65.6\%$ | $-83.6\%$ | $-21.3\%$ |
| Jumper | 3.5 | $-28.6\%$ | $+2.9\%$ | $\mathbf{-42.9}\%$ | $+40,0\%$ | $-40.0\%$ | $-31.4\%$ | $-5.7\%$ |
| Leaper | 0.6 | $-450.0\%$ | $\mathbf{-600.0}\%$ | $-66.7\%$ | $-783,3\%$ | $+0.0\%$ | $+0.0\%$ | $+16.7\%$ |
| Maze | 4.8 | $-27.1\%$ | $-81.3\%$ | $-54.2\%$ | $\mathbf{-137,5}\%$ | $-52.1\%$ | $-83.3\%$ | $-20.8\%$ |
| Miner | 4.3 | $\mathbf{-111.6}\%$ | $-7.0\%$ | $-34.9\%$ | $-88,4\%$ | $-25.6\%$ | $-53.5\%$ | $-30.2\%$ |
| Ninja | 3.1 | $-54.8\%$ | $-64.5\%$ | $-54.8\%$ | $\mathbf{-129,0}\%$ | $-35.5\%$ | $-32.3\%$ | $-29.0\%$ |
| Plunder | 2.8 | $-89.3\%$ | $-10.7\%$ | $-32.1\%$ | $+39,3\%$ | $\mathbf{-103.6}\%$ | $-53.6\%$ | $-50.0\%$ |
| StarPilot | 8.6 | $\mathbf{-143.0}\%$ | $+103.5\%$ | $-64.0\%$ | $+153,5\%$ | $-64.0\%$ | $-81.4\%$ | $-45.3\%$ |

Table 16: Generalization gap versus PPO

| Game | PPO | Mixreg | CLOP |
|---|---|---|---|
| bigfish | $1.7 \pm 4.0$ | $6.6 \pm 1.2$ | $\mathbf{13.1} \pm 4.4$ |
| bossfight | $11.4 \pm 0.2$ | $10.1 \pm 0.6$ | $\mathbf{11.7} \pm 0.2$ |
| caveflyer | $4.6 \pm 0.4$ | $\mathbf{5.2} \pm 0.4$ | $4.7 \pm 0.8$ |
| chaser | $9.4 \pm 0.2$ | $7.7 \pm 0.8$ | $\mathbf{9.9} \pm 0.3$ |
| climber | $6.0 \pm 0.5$ | $\mathbf{7.4} \pm 0.5$ | $6.5 \pm 0.7$ |
| coinrun | $6.9 \pm 0.3$ | $6.6 \pm 0.4$ | $\mathbf{7.2} \pm 0.2$ |
| dodgeball | $4.6 \pm 1.0$ | $8.4 \pm 0.6$ | $\mathbf{10.9} \pm 1.2$ |
| fruitbot | $5.4 \pm 1.2$ | $15.5 \pm 1.1$ | $\mathbf{15.7} \pm 0.4$ |
| heist | $1.2 \pm 0.3$ | $1.5 \pm 0.3$ | $\mathbf{2.9} \pm 0.3$ |
| jumper | $3.0 \pm 0.2$ | $\mathbf{4.0} \pm 0.3$ | $2.9 \pm 0.4$ |
| leaper | $8.0 \pm 0.4$ | $6.7 \pm 1.8$ | $\mathbf{8.5} \pm 0.4$ |
| maze | $2.7 \pm 0.4$ | $\mathbf{4.0} \pm 0.5$ | $2.8 \pm 0.3$ |
| miner | $12.3 \pm 0.2$ | $14.5 \pm 0.9$ | $\mathbf{14.5} \pm 0.5$ |
| ninja | $5.9 \pm 0.4$ | $\mathbf{6.3} \pm 0.6$ | $4.8 \pm 0.1$ |
| plunder | $\mathbf{4.4} \pm 0.3$ | $4.4 \pm 0.6$ | $2.8 \pm 1.0$ |
| starpilot | $4.6 \pm 0.6$ | $7.5 \pm 1.2$ | $\mathbf{8.3} \pm 1.4$ |

Table 17: CLOP and mixreg test performance in hard mode.

and letting the critic overfit the value function is actually beneficial. One could conjecture that this phenomenon of overfitted critic benefiting PPO, is also one reason for the success of DAAC (and IDAAC), since their critic is not constrained to share a common stem of convolutional layers with the actor and is thus free to better fit the values.

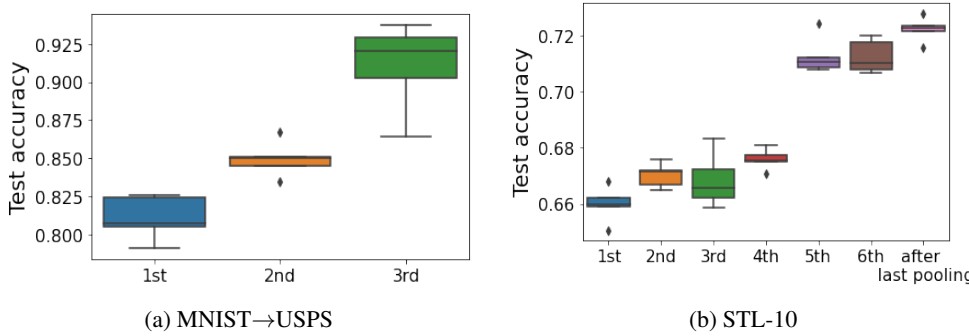

(a) MNIST→USPS

(b) STL-10

Figure 14: Test accuracy for different positions of the CLOP layer

Figure 7 reports the difference in performance between the two versions of PPO, confirming this analysis on a limited set of Procgen games.

## K    WHAT WOULD BE THE EQUIVALENT OF CLOP IN NON-IMAGE-BASED RL?

This short discussion was motivated by comments from the ICLR reviewers.

CLOP is essentially an image-based method that exploits the spatial structure of images. Local permutations are a way of performing local "fuzzying". When such fuzzying is done in the input space it aims at invariance with respect to measurement noise. The rationale behind the CLOP layer is that performing such a fuzzying in feature map space is a principled way to aim at object positions invariance. Consequently, one could see the encoding of the feature maps as isomorphic to a latent space of positional variables. In this latent space, the equivalent of our permutations would correspond (at least in the intention) to the application of a small, centered noise on positional variables, so as to reach positional invariance. From a theoretical perspective, one could interpret this process as a margin maximization problem (the margin being defined in the feature space rather than in the input space). This is an interesting avenue for future research and this short discussion applies more generally to any noise addition process in data augmentation.

