# OpenReview forum: "Local Feature Swapping for Generalization in Reinforcement Learning"
_ICLR.cc/2022/Conference — ICLR 2022 Poster_

### Official Review · Reviewer_VitZ · 2021-10-20

**Correctness:** 4
**Technical Novelty And Significance:** 3
**Empirical Novelty And Significance:** 3
**Recommendation:** 8
**Confidence:** 4

**Main Review:**

Pros:
* As the authors have mentioned, this is a very simple regularization method which achieves good empirical results. From the point of view of judging it alongside other image-based regularization techniques (e.g. cutout, data augmentation) and even RL-specific upgrades, this method seems particularly effective.
* The paper is quite comprehensive in its showing of results (over classification, Procgen + baselines, saliency, visualizations), and via ablations over locality of permutations, show that the method does take advantage of the locality of objects in images strongly. I have a good empirical understanding over the method.

Weaknesses:
* One incredibly small detail that I caught (but I think may matter a lot) in Figure 9, Appendix A.2 is the fact that the CLOP layer isn't used in the image torso shared by both the value and policy heads, but rather only the policy head itself. This makes me somewhat suspicious that one hypothesis is maybe CLOP just affecting the policy's action outputs, leading to random actions during training and thus an exploratory behavior/"data augmentation" that's leading to the higher test performance. This is even more confounded because of the separation between forward passes (for data collection) vs backward passes (where the value head is still affected by CLOP since the image encoder's gradient is affected by CLOP via the policy head). Could you please provide an explanation for why you only used CLOP on the policy head, or add a few extra ablations to refute my proposed hypothesis? For example, what if CLOP is applied at the very end of the image encoder torso, but still shared by both heads? It is not a fatal issue if my proposed hypothesis is true, but analyzing this direction would substantially clarify why CLOP is effective. Apologies if I may be overreacting.

* Since this is a vision-based method, it is naturally difficult to provide more fine grained conceptual analysis on the CLOP regularization method, and thus this paper might not be as impactful to say, someone in the theoretical field as someone in the computer vision field. From the point of view of computer vision, this is a good contribution, but it seems slightly unsatisfying from a theoretical perspective. The method relies on the inductive bias where objects in images are localized, which is true for a large distribution of images. How would this method work on scenarios where this is not the case, especially ones where data other than RGB is used? What would happen if the data also involves e.g. [1] 2D xy coordinates appended into the depth channel, or standard positional embeddings for images sent to a Vision Transformer [2]? It is not obvious to me if CLOP would work here. I wonder if it may be possible to learn the distribution of random permutations during training instead for these non-RGB scenarios.

[1] https://eng.uber.com/coordconv/

[2] https://arxiv.org/abs/2010.11929

**Summary Of The Paper:**

This paper proposes a new regularization scheme, which involves randomly permuting nearby (in terms of spatial locations in the downsized feature tensor) depth blocks. The authors show its effectiveness in supervised learning tasks and establish a SOTA-like result over Procgen generalization, and also provide several empirical ablations and visualizations over the method to understand its inner workings. The paper shows that regularization which involves the locality of objects is particularly effective.

**Summary Of The Review:**

The proposed CLOP method is simple and effective, although I'm afraid it can (to some readers) seem like a (somewhat motivated) "hack", which isn't a huge flaw, but might be limited in impact scope. It is strong to the eyes of an experimentalist, but raises many questions to someone who may be more theoretical.

I am willing to increase the score if the authors answer my main question about the policy head, which made me somewhat alert/suspicious.


EDIT: I have increased my score to an 8 after the rebuttal. The authors have answered my questions thoroughly. Previously, I was concerned that the CLOP method might be a bit too "hacky" (for lack of a better word), but I think the paper's simplicity and strong empirical results may also lead to how we redefine RL generalization. Perhaps Procgen tests too much on "visual generalization" and thus this paper might be providing some evidence for new benchmarks.

---

> ### Author Response · Authors · 2021-11-10
> **Answer to reviewer VitZ**
>
> Thank you for these very insightful comments. Thank you also for the very kind and constructive phrasing you use: we strongly appreciate it and will do our best to address your comments. We report below the changes we are currently making to the paper to address your comments. We plan to upload the updated version of the paper as soon as all results and edits are consolidated, hopefully at the beginning of next week.
>
> ## Why is CLOP used only on the policy head in PPO?
> No need to apologize, we believe the question is very legitimate and hope this answer will clarify things and lift what made you suspicious. Our answer here is two-fold.
> First, PPO is a "true" policy gradient method, contrarily to SAC or TD3 that are approximate value iteration methods which include gradient ascent on the policy to solve the greediness search (the $\arg\max$ operator in the Bellman equation). Because of this, it collects samples at every round to estimate the critic and cannot rely on a persistent replay buffer. Intrinsically, the critic is only used to compute advantage estimates in states that have been encountered during recent exploration. Consequently, there is no need for the critic to actually generalize to unseen states and letting the critic overfit the value function is actually beneficial. We have tried to summarize this into the single sentence "Since the critic’s role is only..." in the "CLOP within PPO" paragraph. We propose to add a paragraph in the appendix to expand this discussion. One could conjecture that this phenomenon of overfitted critic benefiting PPO, is also one reason for the success of DAAC (and later IDAAC), since their critic is not constrained to share a common stem of convolutional layers with the actor and is thus free to better fit the values. Of course, if one wanted to apply CLOP with TD3 or SAC (or even DQN), then the CLOP layer would absolutely need to be included in the critic.
> Secondly, and to confirm the previous argument, we are currently running an alternate version of our experiments where CLOP is applied both on the actor and the critic. We agree with your comment that this would make for an interesting ablation and will post an update here as soon as we have consolidated results.
>
> ## What would be the equivalent of CLOP in non-image-based RL?
> As you noted, CLOP is essentially an image-based method that exploits the spatial structure of images (just like CNNs do, on a more general note). Local permutations are a way of performing local "fuzzying". When such fuzzying is done in the input space it aims at invariance with respect to measurement noise. Our main point is that doing it in feature map space is a principled way to aim at object positions invariance. Consequently, one could see the encoding of the feature maps as isomorphic to a latent space of positional variables. In this latent space, the equivalent of our permutations would correspond (at least in the intention) to the application of a small, centered noise on positional variables, so as to reach positional invariance. From a theoretical perspective, one could interpret this process as a margin maximization problem (the margin being defined in the feature space rather than in the input space). We completely agree this is an interesting avenue for future research but also remark that this applies more generally to any noise addition process in data augmentation.
> One additional strong interest of doing this in a feature map space is that this space is spanned by the successive application of convolutional layers (9 in the IMPALA architecture), and thus is hopefully a good representation of the spatial structure of the image, ie the position of objects or salient groups of pixels. Applying noise to scalar values such as positions begs the question of the noise intensity, which is related to the scaling of each of these variables. In a way, multi-layered CNNs avoid this question altogether by allowing the projection in the latent space to be "learned" by gradient descent.
> We propose to add this discussion as an additional paragraph in the appendix (and refer to it from the main text).

---

> > ### Comment · Reviewer_JpKh · 2021-11-11
> > **CLOP with other RL algorithms?**
> >
> > Thank you for the quick responses.
> >
> > Just jumping on this thread to follow up on VitZ's comment and the author response. Personally, I would be significantly more convinced in CLOP as a general approach for improving RL generalization if it was shown to consistently boost performance with various RL algorithms. This is similar in spirit to the comment many reviewers including myself had in the first round, about evaluating more thoroughly for supervised learning: basically, does CLOP only work with this specific architecture and algorithm instantiation, and for these specific tasks?

---

> > > ### Author Response · Authors · 2021-11-19
> > > **Paper update**
> > >
> > > Thank you to reviewer VitZ for the initial suggestions, and to reviewer JpKh for kicking in on this part of the discussion: it is very much appreciated.
> > >
> > > We have included the specific experiment about applying CLOP on the critic within the results reported in Figure 7 (Section 5.2) of the main text and referred to Section J of the appendix which features a more detailed discussion. As you both mentioned, this is a general question of where to apply CLOP within a network (both in RL and SL), which we also discuss now in Section I of the appendix.
> > >
> > > We also added Section K of the appendix, which is strongly inspired by our answer to your question about the equivalent of CLOP in non-RGB state spaces.
> > >
> > > Due to the limited amount of time, we could not run an additional experiment in RL, using another algorithm. We instead consolidated the SL results (which, we believe, has a broader impact than testing another RL algorithm) and provided a better coverage of results with PPO. We also extended the results by combining IDAAC and CLOP.
> > > Although we agree that more algorithms tested are always better, we also remark that most RL algorithms are approximate dynamic programming methods (most often approximate value iteration methods) whose convergence bounds rely on the accuracy properties of the approximation operator (Scherrer et al, 2015). Consequently, strengthening the results in supervised learning directly affect what one could expect of algorithms such as SAC, TD3 or DQN. This does not constitute a proof per se, but we believe it to be a step in the discussion (and ADP is known to be the place of many interacting phenomena that are often hard to disentangle).
> > >
> > > Scherrer, B., Ghavamzadeh, M., Gabillon, V., Lesner, B., & Geist, M. (2015). Approximate modified policy iteration and its application to the game of Tetris. J. Mach. Learn. Res., 16, 1629-1676.
> > >
> > > We hope that given your initial positive evaluation on this contribution, the improvements that follow our discussion will convince you to lean even more toward acceptance of the paper.

---

> > > > ### Comment · Reviewer_VitZ · 2021-11-21
> > > > **Increased my score**
> > > >
> > > > I have increased my score to an 8 after the rebuttal. The authors have answered my questions thoroughly. Previously, I was concerned that the CLOP method might be a bit too "hacky" (for lack of a better word), but I think the paper's simplicity and strong empirical results may also lead to how we redefine RL generalization. Perhaps Procgen tests too much on "visual generalization" and thus this paper might be providing some evidence for new benchmarks.

---

### Official Review · Reviewer_8Mbh · 2021-10-31

**Correctness:** 3
**Technical Novelty And Significance:** 2
**Empirical Novelty And Significance:** 3
**Recommendation:** 6
**Confidence:** 4

**Main Review:**

**Strengths:**

* The proposed technique is simple and effective.
* The paper is easy to follow.

**Weaknesses:**

* The paper aims at generalization in RL. However, the proposed method seems more like a general data augmentation technique for visual input and CNN. The motivations in section 4 are only loosely connected to RL. For example, "the important information is often spatially scarce and very localized" also applies to image classification.
* It would be better to use the same experiment protocol as in the Procgen paper (full distribution of levels for testing instead of 1000, for hard difficulty). Also, the choice of the three environments (Dodgeball, Miner, Chaser) seems arbitrary. In addition, it would be better to include comparison with other methods that also use hard difficulty such as mixreg.
* The results of IDAAC and CLOP in table 2 seem comparable. Though the improvement may be complementary, it is better to justify with experiments (i.e., IDAAC + CLOP)
* To enable direct comparison, it would be better to compute a normalized score, as suggested by the Procgen paper.
* Why table 3 only includes the comparison of CLOP and PPO? The training and testing results for IDAAC / mixreg / UCB-DraC are available in IDAAC paper.
* To make the results more convincing, it would be better to conduct ablation experiments (Figure 5 and Figure 7) on all environments.

**Summary Of The Paper:**

This paper introduces an augmentation technique: channel-consistent local permutations (CLOP), to help address observational overfitting issue in RL. The proposed method is evaluated on Procgen benchmark and outperforms other methods.

**Summary Of The Review:**

Overall, I think the proposed augmentation technique is simple and effective when compared to original PPO and other data augmentation techniques, but it does not significantly outperforms state-of-the-art IDAAC. I have some concerns on its connections to RL and some experiment details. Therefore, at this moment I lean towards rejection.

---

> ### Author Response · Authors · 2021-11-10
> **Answer to reviewer 8Mbh**
>
> Thank you for these constructive comments. We report below the changes we are currently making to the paper to address your comments and make the contribution more convincing. We plan to upload the updated version of the paper as soon as all results and edits are consolidated, hopefully at the beginning of next week.
>
> ## Is this method for RL or is it more general?
> We agree that CLOP is "task-agnostic" and applies to supervised learning as well as RL. However, we believe full evaluation of CLOP in supervised learning is actually a very ambitious endeavor. It would, for instance, require evaluations on classification tasks, but also image segmentation, or video data. It would also require extensive training on a variety of datasets to provide reliable take-away messages. The reason we present CLOP primarily as a contribution in RL is mainly prudence. We believe claiming a systematically high performance in supervised learning is beyond what is shown by our experimental results. It has become a common practice in ML research to present each contribution as an unprecedented breakthrough and we advocate for humble, factual papers, so we try to apply this to our own contributions. Maybe CLOP has the potential to become a standard in regularization, just as dropout has become; we actually believe it is the case, but to claim this, we would probably need a more extensive study on where and how CLOP should be included in NN architectures, which we reserve for future work. Nevertheless, we will do our best (with limited resources and time) to run the experiments you suggested on ImageNet (and also a comparison with MixUp), for completeness in the current paper.
>
> ## Procgen hard mode
> The experimental protocol in the Procgen paper implies testing on the full distribution of levels, including those used during training. Our choice of using only the testing levels makes the benchmark even harder since no level seen during training is among our testing distribution. We argue our protocol has two benefits. We test only the generalization property, without any influence from the performance on training levels. Also, the levels selected for testing along the training process are precisely the same across algorithms. In order to address your concern, we will add the performance on the full distribution of levels (including the training ones) in the appendix.
> We also added, in the appendix, the full results on the hard mode of Procgen. Please note that these results are consistent across games. There are games where CLOP is actually even better than in the reported cases from the main text of the paper (e.g. bigfish, bossfight, heist, leaper, starpilot). Overall, these results confirm those obtained in the easy mode. We hope this lifts your legitimate doubts that we might have cherry-picked these three environments. We also included the results of MixReg as a comparison baseline as you suggested.
>
> ## Performance assessment and combination with IDAAC
> We agree that the gap between CLOP and IDAAC is rather game-dependent. Nevertheless, CLOP sets a new state of the art on 7 of the Procgen games, it is top 2 on 10 of the 16 games. None of the main competitors brought such an improvement at the moment of release (IDAAC, for example, dominated on 6 games versus its reported competitor). Concerning the specific gap with IDAAC, it is rather game-dependent and can be quite large (eg. Starpilot, Leaper, Chaser, Dodgeball). A major advantage is that CLOP is a conceptually simple method, offering hyperparameter robustness, where the direct competitor (IDAAC) is a lot more convoluted (it separates networks, needs fine hyperparameter tuning, uses the advantage prediction as an auxiliary loss, and introduces and adversarial loss which is prone to learning instabilities). We believe discarding the results of CLOP as "not significant[ly] outperform[ing]" IDAAC is a bit unfair in this context.
> We agree on the proposal to report results on the combination of CLOP and IDAAC. These experiments are currently running and we will post an update here as soon as we have consolidated results.
>
> ## Extending table 3
> Table 3 already has 7 columns and we did not include the other algorithms in the table for readability. We have added them in the appendix and refer to them from the main text.
>
> ## Ablation experiments
> These are currently running on all environments (to complement those on Bigfish, Dodgeball and Chaser) to demonstrate their consistency across environments.

---

> > ### Author Response · Authors · 2021-11-19
> > **Paper update**
> >
> > We believe we have addressed all your expressed concerns and put in a major effort to make the contribution better.
> >
> > Here are the modifications we made to address your concerns:
> >
> > > The paper aims at generalization in RL. However, the proposed method seems more like a general data augmentation technique for visual input and CNN. The motivations in section 4 are only loosely connected to RL. For example, "the important information is often spatially scarce and very localized" also applies to image classification.
> >
> > We have added experiments on Imagenet, STL-10 and Imagewoof, included the comparison with mixup, and enhanced the discussion on generalization both in RL and SL. This now appears in Sections 5.1, 5.2, and appendix D (and also appendix sections E to
> >
> > > It would be better to use the same experiment protocol as in the Procgen paper (full distribution of levels for testing instead of 1000, for hard difficulty). Also, the choice of the three environments (Dodgeball, Miner, Chaser) seems arbitrary. In addition, it would be better to include comparison with other methods that also use hard difficulty such as mixreg.
> >
> > Section G of the appendix (and the corresponding discussion in the main text in Section 5.2) has been enhanced to both explain our experimental protocol, and include the results on the protocol of the Procgen paper. We also added the results on all Procgen games in hard mode, and the comparison with mixreg, which both confirm and strengthen our initial claims.
> >
> > > The results of IDAAC and CLOP in table 2 seem comparable. Though the improvement may be complementary, it is better to justify with experiments (i.e., IDAAC + CLOP)
> >
> > Please see our answer in the previous comment concerning the comparison between the performance of CLOP and IDAAC.
> > We included the combination of IDAAC and CLOP in section E of the appendix along with a discussion.
> >
> > > Why table 3 only includes the comparison of CLOP and PPO? The training and testing results for IDAAC / mixreg / UCB-DraC are available in IDAAC paper.
> >
> > Section F of the appendix now report the comparison with all algorithms (those you ask for, plus all the others mentioned in the paper). It also features a deeper discussion on the evaluation of the generalization gap and its evolution along training.
> >
> > > To make the results more convincing, it would be better to conduct ablation experiments (Figure 5 and Figure 7) on all environments.
> >
> > They are reported in section H of the appendix.
> >
> > We hope these modifications will convince you of the contribution's value and change your overall appreciation of the paper.

---

> > > ### Comment · Reviewer_8Mbh · 2021-11-20
> > > **Feedback to the authors' response**
> > >
> > > Thank authors for the efforts in responding to the comments and running additional experiments. The rebuttal has addressed some of my concerns, and I appreciate the authors adding experiments on supervised tasks and Procgen hard mode. These updates make the paper stronger. Therefore, I decide to raise my score.
> > >
> > > However, I am still not convinced that CLOP outperforms IDAAC. The authors keep emphasizing the number of games where CLOP tops, but the improvement is marginal for most games. I agree that the gap can be game-specific. That's also the reason why I suggest the authors report a single normalized score following the protocol in procgen paper. In addition, I feel the authors are dodging my comment of adding other methods to the comparison in Table 3. The new Table 12 shows the generalization gap itself while Table 3 shows the improvement in the generalization gap. Besides, the results in Table 9 do not support the claim that IDAAC and CLOP are complementary, while the authors still use such claim in Section 5.2 (the paragraph *Comparison with RL regularization methods*).
> > >
> > > Overall, I still slightly lean towards rejection. I agree CLOP is much simpler than IDAAC, but IDAAC offers interesting insights on the connection between decoupling value/policy and the generalization ability, while CLOP looks like yet another regularization technique. Thus I expect a significant empirical improvement over IDAAC.

---

> > > > ### Author Response · Authors · 2021-11-22
> > > > **CLO vs IDAAC, normalized scores, and general thoughts**
> > > >
> > > > Since we still have until tonight to update the paper, we edited the Appendix to better address your concerns. You will find two new paragraphs, 4 new tables and 1 new figure in red, pages 16 to 23.
> > > >
> > > > To be able to report the same table as Table 3, we would need to have a fair baseline.
> > > > Comparing IDAAC to our PPO results seemed a bit like comparing apples and oranges since the network architectures and implementations are different.
> > > > So we initially chose to report the actual generalization gap to let the reader make an informed opinion.
> > > > So, no dodging whatsoever :-).
> > > > We had actually insisted on the difference between performance gap and performance gap gain (last sentence of the first paragraph in the corresponding appendix F).
> > > > We have now included these tables for training levels (Table 14), testing levels (Table 15), and generalization gap (Table 16) to convince you, our PPO results as the baseline for all algorithms. But we're rather cautious regarding their interpretation (e.g. one can get a great generalization gap with a terrible overall performance).
> > > >
> > > > Concerning normalized scores, we added the table with them for the comparison of CLOP and IDAAC (Table 13) and also added histograms of the difference in performance.
> > > > One word of warning here: as we discuss on pages 16-17, the scaling factors in the Procgen paper can actually be quite far from the top performance obtained by any algorithm.
> > > > This tends to play in favor of IDAAC since it shrinks the domination of CLOP on Dodgeball, Bigfish or Starpilot.
> > > > Adding these histograms actually brings more information to the reader: thank you for pointing us in this direction.
> > > > We discuss these results at length in an additional paragraph in Appendix F.
> > > >
> > > > Lastly, although we still wish to put forward the performance of CLOP (and we argue there should be no shame at being on par with state-of-the-art methods, often better, outperforming all others, and reporting results on hard mode), we'd also like to pinpoint the fact that this is not purely a competition about who has the largest number of points (and, even more importantly, it is not just a contest of CLOP vs. IDAAC: there are lots of good performing algorithms beyond IDAAC).
> > > > It is rather a search for insights, as you seem to think too.
> > > > We totally acknowledge and highly value the contribution of IDAAC --- "contributions" (plural) would actually be fairer since IDAAC relies on the combination of actor/critic separation (DAAC) and auxiliary losses.
> > > > We believe CLOP takes a different point of view, somehow more related to the intention of IDAAC's auxiliary losses than to the actor/critic separation, and brings in different insights concerning translation invariance of features.
> > > > We don't mean to be nitpicking but yes, CLOP is yet another regularization technique, just as the adversarial loss of IDAAC was (or the contributions of IBAC-SNI or UCB-DraC); does this make it a minor contribution?
> > > >
> > > > Again, we thank you for insisting on these aspects that make the discussion deeper for the reader and helped us improve the paper.

---

> > > > > ### Comment · Reviewer_8Mbh · 2021-11-22
> > > > > **Feedback**
> > > > >
> > > > > Thank the authors for the response. The normalized score (Table 13) and the histograms (Figure 11) now clearly show the performance gain of CLOP. For this reason, I raise my score.
> > > > >
> > > > > Meanwhile, I have a few comments in response to some of the authors' remarks.
> > > > > * If the authors agree that **gain in generalization gap** does not align with the overall benefits we want, then they should simply get rid of this measure and do not include it in Table 3. It is just not fair to first say in the main content that "CLOP layer systematically improves the generalization gap" based on a table without other methods, then in the appendix say something like "gain in generalization gap does not really count" for a complete table.
> > > > > * To be clear, I am not saying that a good paper  must have the highest number. It would be meaningless to base everything on the performance. But if a method itself does not offer much insights (and it is also not theoretically supported), then all I expect is its empirical performance. If it performs amazingly well, then it is worth exploring why it works so well in follow-up studies. But if it does not bring much improvement, then the community can not benefit a lot from such a method.
> > > > > * The authors argue that the insight here is the translation invariance of features. But most augmentation techniques are concerned with translation invariance, in (slight) different ways. What makes CLOP special? Finally, I do not aim for further debate about the contributions of other papers so let's stop here.
> > > > >
> > > > > In addition, please modify the claim that IDAAC and CLOP are complementary in Section 5.2, as suggested in my previous reply.

---

### Official Review · Reviewer_JpKh · 2021-11-01

**Correctness:** 3
**Technical Novelty And Significance:** 4
**Empirical Novelty And Significance:** 3
**Recommendation:** 8
**Confidence:** 4

**Main Review:**

Strengths:
- A novel technique (to my knowledge) for CNN regularization, a topic that is well-studied for many years now
- Strong experimental design and results, spanning supervised and reinforcement learning settings, particularly, generalization across levels.

Weaknesses and clarification questions:

- The supervised learning results are a bit superficial, and don't evaluate against state-of-the-art approaches like Mixup, and on more standard tasks like the full ImageNet task. I do appreciate that that is not the aim of the authors, but since the proposed method itself doesn't have much to do with RL, and the impact of the paper would be broader if it added more thorough supervised learning experiments too, to establish the proposed approach as the state of the art for CNN regularization in general vision tasks.

- Fundamentally, augmentation is a way of expressing known invariances to "label-preserving transformations" in the data. We normally perform augmentation in the input space because this is where we know the invariances. In a latent feature space many layers deep into the CNN, we don't know what is represented. What does it mean to do augment the data here? How do we know that moving the features won't change the optimal action? For example, in a driving scenario, if CLOP moves a pedestrian in a scene from the pavement onto the road, shouldn't the driving agent respond significantly differently? Have the authors observed systematic ill effects from CLOP in some settings?

- How does the layer to which CLOP is applied affect performance? Besides the prescription of applying it to the last convolutional layer, some empirical results addressing this would be useful.

- The writing is sometimes a bit convoluted. The text in Sec 2 Generalization in RL, I found more confusing than illuminating.

**Summary Of The Paper:**

This submission proposes a simple new regularization technique for convolutional neural networks: "channel-consistent local permutations" (CLOP): at a deep layer in a CNN, it randomly swaps some pixels with their neighbors in the forward pass during training. This is shown to outperform other approaches for regularization, both in supervised learning and in reinforcement learning settings, showing particular benefits for generalizing across levels of procedurally generated games.

**Summary Of The Review:**

An interesting and neat approach to regularize CNNs, motivated by the need to learn task-relevant representations invariant to background and other distractions for visual RL tasks. Strong results, well-designed experiments, and well presented. I am in favor of accepting this paper, but await the author responses and other reviewers' comments to form a stronger opinion.

---

> ### Author Response · Authors · 2021-11-10
> **Answer to reviewer JpKh**
>
> Thank you for these insightful comments. We report below the changes we are currently making to the paper to address your concerns. We plan to upload the updated version of the paper as soon as all results and edits are consolidated, hopefully at the beginning of next week.
>
> ## CLOP at various depths in the network
> Thank you for this great remark. We actually had formulated the same thought after submitting the paper. Overall, we believe this opens a more general research question that is related to how to design neural network architectures to enforce regularization: where and how is it relevant to include CLOP, dropout, batchnorm or other regularization methods? Although this general question is beyond the scope of this paper, we believe assessing the contribution of CLOP at different depths remains an insightful element for the present study. We are currently running the corresponding experiments and will post an update here as soon as they are ready.
>
> ## CLOP in supervised learning
> The reason we present CLOP primarily as a contribution in RL is mainly prudence. Even though we totally agree that CLOP is a "task-agnostic" method, we believe claiming a systematically high performance in supervised learning is beyond what is shown by our experimental results. It has become a common practice in ML research to present each contribution as an unprecedented breakthrough and we advocate for humble, factual papers, so we try to apply this to our own contributions. Maybe CLOP has the potential to become a standard in regularization, just as dropout has become; we actually believe it is the case, but to claim this, we would probably need a more extensive study on where and how CLOP should be included in NN architectures, which we reserve for future work (see previous paragraph). Nevertheless, we will do our best (with limited resources and time) to run the experiments you suggested on ImageNet and the comparison with MixUp, for completeness in the current paper.
>
> ## Ill-effects from CLOP
> Interestingly, we have not observed any of the ill-effects from CLOP that you mention, even though we expected them as well. Our interpretation is as follows. In most cases, local swapping preserves the label: following up on your example, there are a lot more permutations that shift the red light's position slightly, or that move the pedestrian around her position on the sidewalk, than the ones that actually put her on the road. If the different channels of the deepest feature maps actually correspond to high-level features that condition the action or the value (like positions of objects), then it is at this level that we would like to obtain generalization properties. This is somehow complementary to the invariance to image measurement noise that is often sought in data augmentation in the input space. Additionally, for the samples that become ill-labeled due to the application of CLOP, it drives the actor into taking the action in future trajectories, thus collecting the corresponding true reward and correcting the label. We shortly discussed this aspect in the paragraph about the importance of locality in the permutations. We propose to add a paragraph in the Appendix (due to lack of space in the main text) in order to expand this discussion.

---

> > ### Comment · Reviewer_JpKh · 2021-11-11
> > **Explanation of the label correction?**
> >
> > Thank you for the quick response.
> >
> > Not sure I follow the logic for how wrong labels due to CLOP augmentations might get corrected automatically.
> >
> > Separately, if I were to design a simple experiment to see how much positional/spatial information is lost, it might take the form of an object detection style problem where CLOP could force bounding box offsets proportional to the swapping distances.

---

> > > ### Author Response · Authors · 2021-11-19
> > > **Paper update**
> > >
> > > Thank you again for your answers in the discussion!
> > >
> > > Here is a summary of the modifications we made to address your concerns:
> > > - We have included the comparison with mixup in appendix D (and referred to it from Section 5.2). It confirms and strengthens the claims made previously. Applying CLOP on Imagenet also highlighted specific aspects of regularization, which motivated the additional experiments on STL-10 and Imagewoof and the discussion of appendix D. Thank you again for triggering this discussion which, we believe, is a strong improvement on the paper.
> > > - We included a more thorough analysis of the influence of the CLOP layer's position (Section 5.2, last paragraph and appendix I and J). Again, making this explicit helped making stronger points in the paper.
> > > - We tried to rephrase part of Section 2 to clarify it (and more generally tried to incorporate all reviewers' comments on clarity).
> > >
> > > Concerning the question about mislabeling augmented samples, we can simplify our two arguments:
> > > - First, it only concerns a minority of samples. This is because most of the permutations actually swap irrelevant elements of the image (e.g. background elements), and also because most local displacements of objects don't strongly affect the optimal action.
> > > - Secondly, mislabeling is less of a problem in RL than in SL because of the exploration process. In SL, mislabeling a sample might induce a bias in the prediction function (this is also the case for any method that introduces inductive biases). In RL, the exploration process will lead the agent into trying out the actions that have been enforced in the policy and exploring around them. So exploration itself might bring corrected samples into the replay buffer.
> > > Does this clarify why the inductive bias of CLOP is likely to be a reasonable data augmentation choice?
> > >
> > > Overall, we acknowledge the very positive impact your comments had on our paper. We hope that given your initial positive evaluation on this contribution, the improvements that follow our discussion will convince you to lean even more toward acceptance of the paper.

---

> > > > ### Comment · Reviewer_JpKh · 2021-11-23
> > > > **Update**
> > > >
> > > > Thank you for the responses. The new supervised learning results move my score to 8.
> > > >
> > > > The "CLOP+RL benefits from label correction" argument as I understand it is: moving location-critical items once in a while is okay (as CLOP does), because those same or very similar states will likely be encountered again.
> > > >
> > > > This is true for deterministic environments / environments with a small state space, but not really true for more realistic settings: like a driving setting with other agents etc. where the same state may never be encountered again. I'd recommend avoiding such statements in the final paper, or making them with more careful qualifications.

---

### Official Review · Reviewer_2gnP · 2021-11-02

**Correctness:** 3
**Technical Novelty And Significance:** 2
**Empirical Novelty And Significance:** 3
**Recommendation:** 8
**Confidence:** 4

**Main Review:**

STRENGTHS

I want to start by acknowledging that the authors provided a version of the code used in their experiments which was very easy to navigate through and understand.

The main strength of the regularization method introduced in this work is in its simplicity. Swapping dimensions of an intermediate feature map while keeping channel-wise consistency does not require heavily engineered functions  and is potentially applicable besides images to other visual inputs like videos. The only introduced parameter in the method, alpha, is the one that controls the probability of randomly swapping an element of the feature map with a random neighbour. Generalization seems not to be sensitive to the value of alpha as long as it is non-zero, implying the lack of necessity for heavy hyperparameter tuning.

Another important contribution of this work is in the improved generalization performance obtained in a supervised learning as well as a RL setup. In the former, CLOP improves accuracy on two vision dataset. For the latter, the authors validate CLOP on a set of procedurally generated RL environments. Compared to other commonly used regularizations, their method ranked first or second in terms of generalization performance on 10 out of 16 environments.


WEAKNESSES


Throughout the paper, CLOP computational complexity is defined as negligible without any explanation. Does CLOP have to process each element of the intermediate feature map, thus scaling with a quadratic cost assuming a constant cost for a swap operation, and is considered a negligible overhead given that features maps usually have low dimensions?
Without a proper cost analysis it is difficult to judge the tradeoff of using CLOP and I am not in favor of accepting this work without any discussion of the reportedly negligible cost.

I agree with the claim that the type of spatial regularization that CLOP tries to inject works at higher layers of neural networks. However, I’m questioning the robustness of using CLOP across different layers. Someone wishing to use CLOP in an RL setup would greatly benefit from an analysis of its performance across different parts of the model. Did the authors try applying CLOP on a deep layer that is not the last one in the model architecture?
With evidence of the robustness of the proposed method I am in favor of vouching for the acceptance of the paper.

In this supervised learning results (section 5.1, table 1), CLOP seems to be similarly beneficial when using MNIST as well as Imagenette, with the latter being a visually richer dataset.
Moreover, Imagenet(te) pictures are iconic, with the target object often clearly visible at the center of the image. For this reason, being able to achieve things like background invariance, possibly induced by CLOP, should be more beneficial for Imagenette compared to MNIST/USPS. This makes me wonder about the scalability of CLOP to richer vision datasets. Did the authors try to validate their results on a larger-scale experiment (e.g. ImageNet or CIFAR100). While this is not a fundamental weakness, I think the paper would benefit by the inclusion of this experiment. On this topic, why is the paper presented mainly as a reinforcement learning regularization method if supervised learning results are also reported?

----

Minor Suggestions or enhancements:


The generalization taxonomy in section 2 is rather confusing, especially given that the paper later focuses on the “simplest” type of generalization among those presented. This is the problem of generalizing within a single MDP and is defined as observational overfitting (OO) at the end of section 3. However, when first introduced quoting Song et al. (2019), OO is defined as the generalization to a distribution over MDPs and, even if presented in the text, it is not clear how the generalization phenomenon studied in this work departs from Song’s formulation.

In the code, I noticed that the Imagenette experiments used a stochastic flipping function to preprocess the input. This should be mentioned in the paper, possibly in a footnote.

While I appreciate the authors explicitly reporting standard deviation values in their results, I did not find mentions of the number of repeated runs for each experiment until table 6 in the appendix. I suggest the authors briefly report this in the main text or point to the appendix for a better interpretation of the presented results.

In the “observational overfitting in RL paragraph”, OO is mentioned (“Therefore, preventing OO in RL remains the ability etc.)” without being properly defined until later in the text (“Song et al. call the problem of OO etc.”).

There could be more consistency in the choice image encoding: in section 4, an image is described as C x H x W, then in section 5.2 an image is described as 64 x 64 x 3.

Across plots and tables, except for the supervised learning results, the “no regularization” setup is defined as PPO and compares to CLOP and several others regularization/augmentation strategies. I find it clearer to define said setup as something like “no regularization” or “plain setup” or something of the sort.

I was able to grasp the main message behind table 3 but I found the results on the CLOP gap slightly confusing. It might be worth mentioning in the caption how the gap is computed.


**Summary Of The Paper:**

In this work, a novel regularization method for deep neural networks is introduced. By locally swapping dimensions of intermediate feature maps, the authors report generalization improvements in supervised learning and on several reinforcement learning benchmarks. The method is evaluated against several other common regularization techniques and was found to lead to better results.

**Summary Of The Review:**

Overall, the paper is clearly written and easy to follow and I find the idea behind CLOP simple and effective. However, concerns regarding scalability and robustness of the proposed method should be addressed by the authors. I am happy to raise my score and vote for the inclusion of this work at ICLR provided that authors report evidence of a deeper empirical analysis.


===============

UPDATE: I raised my score from 5 to 8. I think that the analysis conducted by the authors have drastically improved the paper. I believe it meets the criteria to be included in the conference. The main contributions of this work are the introduction of a new regularization method (CLOP) for image-based RL and supervised learning tasks as well as a detailed empirical analysis and evaluation on several standard benchmarks.

---

> ### Author Response · Authors · 2021-11-10
> **Answer to reviewer 2gnP**
>
> Thank you for this insightful review. We report below the changes we are currently making to the paper to address your comments. We plan to upload the updated version of the paper as soon as all results and edits are consolidated, hopefully at the beginning of next week.
>
> ## Computational cost of the CLOP layer
> Indeed, CLOP has to process each element of the intermediate feature map, but only width and height-wise. this results in a $O(H'W')$ cost, where $H'$ and $W'$ are the height and width of the feature maps, which can be much smaller than that of the input image. An important feature is the independence on the channel depth, which can grow significantly in deep layers. Comparatively, methods that perform data augmentation in the input space generally suffer from complexities in $O(CHW)$ (for input images of size $C\times H\times W$), and sometimes more for costly augmentation methods like rotations. We will edit the paper in order to be more precise on these aspects. We will also include a report of the running time of a PPO training with and without CLOP, in order to strengthen the claim of negligible overhead cost. Does this address your concern?
>
> ## CLOP at various depths in the network
> Thank you for this great remark. We had formulated the same thought after submitting the paper. Overall, we believe this opens a more general research question, related to how to design neural network architectures to enforce regularization: where and how is it relevant to include CLOP, dropout, batchnorm or other regularization methods? Although this general question is beyond the scope of this paper, we believe assessing the contribution of CLOP at different depths remains an insightful element for the present study. We are currently running the corresponding experiments.
>
> ## CLOP in supervised learning
> The reason we present CLOP primarily as a contribution in RL is mainly prudence. Even though we totally agree that CLOP is a "task-agnostic" method, we believe claiming a systematically high performance in supervised learning is beyond what is shown by our experimental results. It has become a common practice in ML research to present each contribution as an unprecedented breakthrough and we advocate for humble, factual papers, so we try to apply this to our own contributions. Maybe CLOP has the potential to become a standard in regularization, just as dropout has become; we actually believe it is the case, but to claim this, we would probably need the extensive study mentioned in the previous paragraph, on where and how CLOP should be included in NN architectures, that we reserve for future work. Nevertheless, we will do our best (with limited resources and time) to run the experiments you suggested on ImageNet (and also a comparison with MixUp), for completeness in the current paper. However, we already formulate the conjecture that the sample density of ImageNet can actually cancel the need for data augmentation altogether. Also, CIFAR-100 is also very "iconic" (the images are scaled and centered) so it might not be the most illustrative dataset to show "the scalability of CLOP to richer vision datasets", for this reason we focus on ImageNet.
>
> ## Generalization taxonomy
> Song et al. introduce a distribution over MDPs and indicate that observational overfitting falls into this category. We argue this might be inexact. Robustness, in expectation, over a distribution of MDPs is what domain randomization (for instance) does. Robustness, in the worse case, over a set of MDPs is what is tackled by robust MDPs. Observational overfitting does not really involve a distribution over MDPs (nor a set of MDPs): it involves a single MDP. To convince oneself of this idea, one can consider a given state and a given action in any Procgen game; there is a unique distribution over next states, not a set of distributions or a distribution over distributions. The confusion might come from the fact that a level is procedurally generated from parameters, which can be drawn from a certain distribution. But this does not define a set of MDPs, it defines a distribution over initial and explored states for a single MDP. In turn, this partial coverage of the explored states is a confounding factor that makes the policy and value function depend on variables that do not allow generalization across states. Consequently, observational overfitting shouldn't be defined as the ability to perform well across MDPs, but rather the ability to perform well across states in a unique, very partially explored MDP.
> Does this reformulation clarify things and should we include it more clearly in the paper?
>
> ## The Imagenette experiments used a stochastic flipping function to preprocess the input
> You are right, we will add it to the paper.
>
> ## Formatting
> We have made consistent the image encoding format and included your remarks concerning the number of runs per experiment, the first mention of OO and the formatting of figures.

---

> > ### Comment · Reviewer_2gnP · 2021-11-12
> > **Response**
> >
> > Thanks for your quick reply on the review and for integrating my comments on formatting and the augmentation for the imagenette experiment.
> >
> > ### Computational cost
> > I find the response, along with the explanation of building the list of random index pairs from the response to reviewer rG3k, convincing. Rather than absolute running time maybe input per seconds on the same machine could make it for a fair comparison between PPO with and without CLOP? Regardless of the one you chose I think this is a nice addition to the paper.
> >
> > ### CLOP at various depths in the network
> > I would say that a paper that introduces a new regularization method should contribute to the discussion on "how to design neural network architectures to enforce regularization". In the end, your arguments with other reviewers about the locality-preserving spatial invariance that CLOP seems to bring is touching upon this topic. Assessing whether CLOP works when applied at a different layer complements the interesting empirical results obtained on Procgen. I'm curious to see the outcome of your current experiments.
> >
> > ### CLOP in supervised learning
> > I appreciate your prudence but then I'm not sure what to make of the supervised learning results from which I cannot infer much relevant information given the controlled data that they used. My remark was that if some experiments were to be included in the paper (and I do not think that it is a necessary requirement, but I believe they make the paper claims stronger) they should be more comprehensive.
> >
> > ### Generalization taxonomy
> > Thanks for your explanation, it is indeed very clear. I'm then wondering if it is necessary to borrow the terminology from Song et al while changing its meaning. In the end, your work studies a type of generalization that is easy to understand and define, I'm not sure the connection to Song et al. is fundamental. This is not a necessary change but I'm wondering whether it could make the writing clearer.

---

> > > ### Author Response · Authors · 2021-11-19
> > > **Paper update**
> > >
> > > Thank you again for your answers in the discussion!
> > >
> > > Here is a summary of the modifications we made to address your concerns:
> > > - Computational complexity: the discussion has been included as appendix C (page 14) (there is a reference from Section 5.2).
> > > - CLOP at various depths in the network is now discussed in Section 5.2 and appendix I (page 20).
> > > - CLOP in supervised learning. Thank you again for these comments, we believe they have brought much depths to the discussion. There is a new reference to appendix D (page 15) from Section 5.2, the overall discussion is enriched by the findings from the additional experiments.
> > > - All other rephrasing or clarification suggestions you made were taken into account and appear within the text of the main paper (in blue).
> > >
> > > We hope the analysis is now deep enough to convince you to raise your score and acknowledge the very positive impact your comments had on our contribution.

---

> > > > ### Comment · Reviewer_2gnP · 2021-11-23
> > > > **Score update**
> > > >
> > > > Thank you for addressing my concerns and provide a deep investigation of several aspects related to CLOP. I believe this work offers interesting empirical insight on image-based regularization techniques in RL and supervised learning. I decided to raise my score from 5 to 8.

---

### Official Review · Reviewer_rG3k · 2021-11-02

**Correctness:** 4
**Technical Novelty And Significance:** 3
**Empirical Novelty And Significance:** 3
**Recommendation:** 8
**Confidence:** 3

**Main Review:**

## Strengths
- The empirical results are comprehensive with great ablation studies. The presentation of the method and the intuition behind it is clear and easy to follow.

## Weaknesses
- Presentation issues
  - The acronyms for the baselines in Table 2 need to be connected to the paper references.
  - Figure 1: it is unclear to me how many pairs of $(h, w)$ are being sampled from $P$. As far as I could tell, this piece of information is also not provided in the rest of the paper. By looking at Figure 2, it seems likely that the for loop is looping over all spatial location of the feature map. It would be good to get clarification from the authors on this.

- Weak experimental results
  - Compared to previous baselines, the proposed method is only better on 7 out of 16 Procgen games. IDAAC is quite competitive as it performs the best on 4 of 16 Procgen games. The gap is even smaller when we are looking at the number of tasks that these approaches get top 2 on. IDAAC is top 2 for 9 out of 16 games and CLOP is top 2 for 10 out of 16 games.
  - The authors mentioned that "Compared to these methods, the CLOP layer offers a direct and easy way to augment the RL agent’s ability to generalize to unseen environments, with the benefit of being entirely complementary with each of them." The combination of the CLOP layer and the previous approach could potentially be quite promising but unfortunately there is no experiment that demonstrates whether CLOP can *actually* bring benefits to the previous approaches.

## Other comments
- In Section 3 last paragraph, the authors stated that "Our method approaches the augmentation of data by noise injection at the feature level, thus avoiding the computationally costly operations of high-dimensional image transformation in regular data augmentation" - From my understanding, data augmentations usually cost a negligible amount of time compared to running the image through the network itself.



**Summary Of The Paper:**

The paper introduces the Channel-consistent LOcal Permutations (CLOP), which randomly swap adjacent features spatially after the last convolution layer in a neural network. Empirical evaluations show that CLOP can improve the generalization of convolutional networks on both supervised and reinforcement learning settings.

**Summary Of The Review:**

The empirical evaluations of the paper are thorough and informative. Although the experimental results are not the strongest (marginal improvements over baselines), they do reflect some effectiveness of the approach despite its simplicity. Because of that, I believe the paper deserves a spot in the conference.

=======================================================================
UPDATE:

Thanks the authors for their response. All of my concerns have been thoroughly addressed. Even though I still think that the improvements of CLOP over previous baselines are not that large, the insights provided by additional experiments and empirical analyses can be quite valuable to the community. Because of that, I have raised my score to 8.

---

> ### Author Response · Authors · 2021-11-10
> **Answer to reviewer rG3k**
>
> Thank you for helping us improve the paper with these accurate remarks. We report below the changes we are currently making to the paper to address your comments. We plan to upload the updated version of the paper as soon as all results and edits are consolidated, hopefully at the beginning of next week.
>
> ## How many pairs $(h,w)$ are sampled from $P$?
> We build a list of indices and shuffle it, then take all $(h,w)$ pairs, one by one. So all pairs are considered for swapping, taken in a random order (to avoid the side effect of taking them in a particular order across the image). The associated complexity is thus $O(H'W')$, with $H'$ (resp. $W'$) the height (resp. the width) of the feature map. We did not include a "shuffle" function for mathematical rigor: $P$ is a set, there is no particular ordering of elements. If you still believe the phrasing "drawn randomly without replacement" is confusing, we could transform $P$ into a vector and add a "shuffle" function.
>
> ## Experimental results
> We are a bit surprised by the qualificatives "weak" and "marginal improvements" on the performance reported. CLOP sets a new state of the art on 7 of the Procgen games, it is top 2 on 10 of the 16 games. None of the main competitors brought such an improvement at the moment of release (IDAAC, for example, dominated on 6 games versus its reported competitor). Concerning the specific gap with IDAAC, it is rather game-dependent and can be quite large (eg. StartPilot, Leaper, Chaser, Dodgeball). A major advantage is that CLOP is a conceptually simple method, offering hyperparameter robustness, where the direct competitor (IDAAC) is a lot more convoluted (it separates networks, needs fine hyperparameter tuning, uses the advantage prediction as an auxiliary loss, and introduces an adversarial loss which is prone to learning instabilities). We believe calling the results "weak" is a bit unfair in this context. Of course we take very seriously all your questions and concerns, and try to address them in detail; we simply thought it was legitimate to share this feeling of surprise after reading the review.
>
> ## Combination of CLOP and IDAAC
> We agree on the proposal to report results on the combination of CLOP and IDAAC. These experiments are currently running and we will post an update here as soon as we have consolidated results.
>
> ## Computational cost of data augmentation
> While some data augmentation methods incur a very small cost (CLOP being one of them), some others can become non-negligible compared to running the image through the network (rotations for instance). We edited the paper to be more precise on this point. One additional advantage of CLOP is its complexity in $O(H'W')$ in the feature space (where the feature map's height $H'$ and width $W'$ can be much smaller than that of the input image), compared to the $O(CHW)$ complexity of most data augmentation methods in input space (for input images of size $C\times H\times W$). We also edited the paper to include this more accurate phrasing.
>
> ## Undefined algorithms' acronyms and missing references
> All apologies for this mistake. It was a side effect of late paper editing. References are back in due place.

---

> > ### Author Response · Authors · 2021-11-19
> > **Paper update**
> >
> > Here are the modifications we made to address your concerns
> >
> > > Presentation issues
> > >    The acronyms for the baselines in Table 2 need to be connected to the paper references.
> >
> > They are back in the paper, page 7, paragraph on "Comparison with RL regularization methods".
> >
> > > Figure 1: it is unclear to me how many pairs of $(h,w)$ are being sampled from $P$ [...]
> >
> > We updated the algorithm in Figure 1, page 5, according to our proposal in the previous comment.
> >
> > > Weak experimental results
> > >    Compared to previous baselines, the proposed method is only better on 7 out of 16 Procgen games. IDAAC is quite competitive as it erforms the best on 4 of 16 Procgen games. The gap is even smaller when we are looking at the number of tasks that these approaches get top 2 on. IDAAC is top 2 for 9 out of 16 games and CLOP is top 2 for 10 out of 16 games.
> >
> > Please see the elements we brought in the previous comment, along with the more detailed experimental results from section 5.2 (pages 6-9) and from Appendices D-G (pages 15-20).
> >
> > > The authors mentioned that "Compared to these methods, the CLOP layer offers a direct and easy way to augment the RL agent’s ability to generalize to unseen environments, with the benefit of being entirely complementary with each of them." The combination of the CLOP layer and the previous approach could potentially be quite promising but unfortunately there is no experiment that demonstrates whether CLOP can actually bring benefits to the previous approaches.
> >
> > Please see  the elements we brought in the previous comment and the results in Appendix E (page 15).
> >
> > > In Section 3 last paragraph, the authors stated that "Our method approaches the augmentation of data by noise injection at the feature level, thus avoiding the computationally costly operations of high-dimensional image transformation in regular data augmentation" - From my understanding, data augmentations usually cost a negligible amount of time compared to running the image through the network itself.
> >
> > Please see the discussion in Appendix C (page 14).
> >
> > All new Appendix sections are referred to from the main text of the paper to enhance the overall discussion and contribution, in Section 5.2.

---

### Public Comment · ~Rishabh_Agarwal2 · 2021-11-09
**Suggestion for reliable evaluation**

Hi authors,

The results table seemed somewhat overwhelming due to mixed results across different games.  Furthermore, we found that the a number of prior reported improvements were only 50-70% likely [1].

Maybe reporting aggregate metrics like Interquartile mean / mean with confidence intervals or metrics like probability of improvement over prior methods might make the results more reliable. Also, performance profiles might serve as a additional visualization in addition to the table. You can easily do so using the library at https://github.com/google-research/rliable or the [colab notebook](https://bit.ly/statistical_precipice_colab).

[1] Agarwal, R., Schwarzer, M., Castro, P.S., Courville, A. and Bellemare, M.G., 2021. Deep reinforcement learning at the edge of the statistical precipice. In NeurIPS.

---

> ### Author Response · Authors · 2021-11-10
> **Thank you for the suggestion**
>
> Dear Rishabh,
> thank you for pointing us towards your paper. We are looking into it right now, if possible we will do our best to follow your recommendation.

---

### Author Response · Authors · 2021-11-19
**Paper update**

We thank all the reviewers for their valuable feedback. We have put in a major effort to address all their comments, questions and concerns and we believe it has brought the paper to a much better level. This has strengthened the results and the claims made in the paper.

All major modifications in the pdf file have been highlighted in blue in order to ease the reading. We will also update to supplementary material in the next days to provide all the code and results for the additional experiments.

Major changes (details in individual answers to the reviewers below) as per the reviewers requests :
- Supervised learning experiments on Imagenet, STL-10, Imagewoof + a discussion on the mechanisms of CLOP + comparison with mixup.
- Full reinforcement learning results on Procgen, hard and easy mode. Detailed generalization gap reduction results.
- Ablation studies on all Procgen games + impact of the CLOP layer's position in the network (depth-wise and on the critic).
- New discussion and empirical evaluation on the computational cost.
- Combination of CLOP and IDAAC and full discussion of results.
- Phrasing clarifications requested by reviewers.

---

### Decision · Program_Chairs · 2022-01-20

**Decision:**

Accept (Poster)

**Comment:**

This paper presents a novel regularization technique for CNNs based on swapping feature vectors in the final layer. It is demonstrated that this simple technique helps with generalization in supervised learning and RL with image inputs.
Following the author rebuttal, all reviewers agreed that the simplicity of this method and the nice empirical performance it obtains is important to report to the community. In this respect, I agree with the reviewers, and recommend acceptance.

One important issue that came up during the discussion is how much this work is related to RL, and the authors SL experiments helped to put the contribution in a broader context. Indeed, one way to see the results of this work is that if such performance improvement is obtained in the Procgen benchmark with just image-based regularization, perhaps this benchmark is not very suitable for studying generalization in RL (where we expect that more sophisticated techniques would be required). In addition, I can think of RL domains (e.g., Tetris, which was mentioned in the discussion) where I would not expect the proposed method to help. It would be good if the authors discuss these issues in some capacity in their final version.

Please take all reviewer comments into account when preparing the final version.